# Critical Evaluation of Current Hypotheses for the Pathogenesis of Hypertrophic Cardiomyopathy

**DOI:** 10.3390/ijms23042195

**Published:** 2022-02-16

**Authors:** Marko Ušaj, Luisa Moretto, Alf Månsson

**Affiliations:** Department of Chemistry and Biomedical Sciences, Faculty of Health and Life Sciences, Linnaeus University, SE-39182 Kalmar, Sweden; marko.usaj@lnu.se (M.U.); luisa.moretto@lnu.se (L.M.)

**Keywords:** hypertrophic cardiomyopathy, hypocontractility, hypercontractility, non-uniformity, hierarchical organization

## Abstract

Hereditary hypertrophic cardiomyopathy (HCM), due to mutations in sarcomere proteins, occurs in more than 1/500 individuals and is the leading cause of sudden cardiac death in young people. The clinical course exhibits appreciable variability. However, typically, heart morphology and function are normal at birth, with pathological remodeling developing over years to decades, leading to a phenotype characterized by asymmetric ventricular hypertrophy, scattered fibrosis and myofibrillar/cellular disarray with ultimate mechanical heart failure and/or severe arrhythmias. The identity of the primary mutation-induced changes in sarcomere function and how they trigger debilitating remodeling are poorly understood. Support for the importance of mutation-induced hypercontractility, e.g., increased calcium sensitivity and/or increased power output, has been strengthened in recent years. However, other ideas that mutation-induced hypocontractility or non-uniformities with contractile instabilities, instead, constitute primary triggers cannot yet be discarded. Here, we review evidence for and criticism against the mentioned hypotheses. In this process, we find support for previous ideas that inefficient energy usage and a blunted Frank–Starling mechanism have central roles in pathogenesis, although presumably representing effects secondary to the primary mutation-induced changes. While first trying to reconcile apparently diverging evidence for the different hypotheses in one unified model, we also identify key remaining questions and suggest how experimental systems that are built around isolated primarily expressed proteins could be useful.

## 1. Introduction

Hereditary hypertrophic cardiomyopathy began to attract appreciable attention in the mid-20th century [1] but it was not until the 1990s that mutations in cardiac sarcomere proteins were found to be causative [1,2,3,4]. The disease affects more than 1 in 500 individuals and is a leading cause of sudden cardiac death in young people [5,6,7]. The disease is due to mutations in one of several different genes out of which the most common are *MYH7* for the ventricular β-myosin heavy chain (β-MHC; 30–40%) and *MBPC3* for the cardiac myosin binding protein C (cMyBP-C; 30–40%). Moreover, the genes for cardiac tropomyosin (Tm; gene: *TPM1*; ~5%), cardiac troponin T (TnT; gene: *TNNT2*; ~5%) and troponin I (TnI; gene: *TNNI3*; 1–5%) are affected, as well as the genes for the myosin regulatory (RLC) and essential light chains (ELC; genes, *MYL2* and *MYL3*; 1–5%) [6]. Mutations in other genes for sarcomere proteins also contribute but are rare. This includes the *ACTC1* gene for cardiac α-actin [6] that is of particular interest because of a high degree of conservation of actin through evolution [8].

Patients with HCM mutations are usually asymptomatic at birth and develop the characteristic disease phenotype over a few years up to several decades, in some cases leading to a severe clinical condition with heart failure, arrhythmias and/or sudden cardiac death [9]. During the pathologic remodeling, changes in cell signaling and other early secondary effects [10] lead to typical histologic/anatomic hallmarks of the disease. These include asymmetric hypertrophy (often in different parts of the interventricular septum), myofibrillar/cellular disarray and scattered fibrosis (e.g., reviewed in References [7,11,12,13]). The hypertrophy is reflected in increased cell size, i.e., hypertrophy of individual cardiomyocytes. Despite the apparently simple causative change behind familial HCM, i.e., single point mutations in a sarcomere protein, several issues remain unclear. For instance, there is ongoing debate about the identity of the primary mutation-induced functional changes that initiate the remodeling. Furthermore, whereas several authors emphasize the importance of common mechanisms for all genes and mutations, others put greater emphasis on gene- and mutation-specific effects. Finally, a multitude of non-genetic factors seem to be important [7].

We here critically review three hypotheses for the primary mutation-induced change behind the pathogenesis in HCM (Table 1). A hypothesis that has received appreciable attention recently is the idea that hypercontractility, as a direct consequence of the mutations, is a key initiator of pathologic remodeling [14,15,16]. This functional change is broadly characterized by increased power output and inefficient energy usage [16], but it also includes features such as slow/incomplete relaxation associated with increased calcium sensitivity of contraction and diastolic dysfunction. The first indication of the feasibility of this hypothesis can be traced back to a review paper in 2003 [11] and then with a further detailed statement in the recent decade [15,16]. Although the hypercontractility hypothesis has quite strong support in the literature, it is not unequivocally supported. For instance, some studies (e.g., References [17,18,19]) open for hypocontractility as an initiating event, suggesting an alternative “hypocontractility hypothesis” (see formulation and critique in Reference [11]). On the other hand, the hypercontractility hypothesis seems to gain support from a recent clinical trial [20] (see more below), showing effectiveness of the drug mavacamten, which acts primarily by reducing force output through the inhibition of actomyosin interactions. One, characteristic that has been associated with hypercontractility in HCM patients is poor energy efficiency of the cardiac contraction [11,21,22,23,24]. This could be a consequence of hypercontractility on the cellular and molecular level due to higher actomyosin cross-bridge cycling rate, but there are also alternative possibilities related to the other hypotheses listed in Table 1.

Finally, according to the “non-uniformity hypothesis” or “contractile imbalance hypothesis” (suggested by several findings [28,33,34,35,36,37]), the main reason for pathogenic remodeling would be non-uniformities in contractile properties among cardiac cells [33,34], possibly exacerbated by mechanical instabilities [36]. This “non-uniformity hypothesis” was inspired by experimental studies that showed increased variability compared to a control group in calcium sensitivity between different single slow muscle fibers carrying β-MHC HCM mutations [28]. Another key finding [38] was the demonstration in model studies that rather small changes in parameter values caused mechanical instabilities in the force–velocity relationship with more than one stable velocity at a given load. One basis for variability in contractile function among cardiomyocytes was suggested to be varying expression levels of wild-type and mutated protein, and evidence to support this idea has later been presented for both the MYH7 and the MBPC3 genes [33,34,37,39,40]. However, other possible reasons for varying contractile function between cells in HCM have been proposed (e.g., References [35,36]), as further considered below.

While being aware of gene- and mutation-specific effects [24,41,42], as well as the importance of complex non-genetic factors and heterogeneities [7,43,44], we here critically review the three major hypotheses mentioned above (hypercontractility, hypocontractility and non-uniformity) for common mechanisms in the pathogenesis of HCM. This focus of the review necessarily means that we do not consider all facets of this complex disease. For a broader and more general insight, we refer the reader to several recent reviews [7,10,14,15,21,24,39,41,45,46,47,48,49] and other studies [13,50,51] that cover the subject from different perspectives. Below, we first more precisely define the different hypotheses. Then we consider each of them in relation to available evidence, with particular emphasis on experiments using preparations of human origin. We also develop the non-uniformity hypothesis further and discuss the possibility that one or more of the key hypotheses may be integrated into one unified model. Finally, we make the case for the use of isolated muscle proteins in the search for key primary initiators of cardiac remodeling in HCM.

## 2. General Definitions and Background

### 2.1. The Myocardial Cells and Their Contractile Machinery

The striated muscles include skeletal and cardiac muscles. In contrast to skeletal muscle, the cardiac muscle is not under voluntary control, and the cells have a branching, rather than cylindrical elongated, structure. Both the skeletal muscle fibers (cells) and the cardiomyocytes are packed with myofibrils that can be subdivided into sarcomeres (Figure 1), the smallest contractile units and the basis for the striated appearance of these muscles. The myofilaments that make up the sarcomere are the thin filaments, which are mostly made up of actin, and the thick filaments, which are mostly composed of myosin. In addition, there are several accessory and regulatory proteins. Tropomyosin and the troponin complex (with subunits T, C and I) in the thin filaments play prominent roles in the regulation of contraction by calcium (Figure 1). Important regulatory/accessory proteins in the thick filaments include cMyBP-C but also titin that bridges the thin and thick filaments.

Myosin (Figure 1C) is a heterohexameric protein with two heavy chains, two essential light chains (ELC) and two regulatory light chains (RLC). The molecule can be subdivided into a globular catalytic domain (head), with actin binding site and catalytic site, a tail (S2 + LMM in Figure 1C) where the two α-helical C-terminals of each heavy chain wind around each other in a coiled-coil and an α-helical neck region between head and tail. The ELCs and RLCs bind to and stabilize the neck regions (Figure 1C). The cross-bridge cycle, where myosin interacts cyclically with binding sites on actin, powered by the turnover of adenosine triphosphate (ATP), is the basis for production of force and shortening [52].

### 2.2. Primary vs. Secondary Mutation Effects and Cardiac Remodeling

On the protein level, a mutation of the genetic code either (i) has no effect on the amino acid sequence, (ii) changes the amino acid sequence without functional effects or (iii) changes the amino acid sequence with more or less severe functional consequences [53]. In the latter case, either the effect is so severe that it is incompatible with life or it has the potential to cause a disease. The severity of changes in protein function depends on location of the amino acid within the protein and the difference in physical properties between the mutated and wild-type amino acid (i.e., positive vs. negative charge, polar vs. non-polar, etc.). The resulting mutated protein can have a gain- or a loss-of-function, varying from minor to severe. Loss-of-function can either imply that less protein is produced or that the protein activity has been compromised. Gain-of-function indicates that the functionality is altered, providing abnormal activity of the mutated protein [54]. The “poison peptide effect”, a gain-of-function mechanism where the mutated protein is incorporated correctly and alters the functionality of the sarcomere [55], is mostly associated with missense mutations in sarcomeric proteins [2,56]. On the other hand, truncating and frameshift mutations do not get incorporated into the sarcomere structure, and the sarcomeric malfunction is due to the lack of functional protein (loss-of-function). The so-called “haplo-insufficiency” or “null-allele” mechanism is mostly associated with mutations in cMyBP-C [57,58]. Overall, the fraction and the location of mutant protein in the sarcomere structure, with the consequent functional changes of the mutation, contribute to the severity of the disease and pathogenic phenotype [59,60].

There are also several non-genetic factors that contribute to the development of the disease over time, such as changes in energy production and demands, accumulation of toxic side products and epigenetic factors [61]. The degree of incorporation of mutated protein into the sarcomere and the correct differentiation and structure of the cells seem to be mutation dependent varying from normal incorporation to early misalignments [62,63].

Because the sarcomere proteins work together to produce force and shortening, the primary effect of any sarcomere protein mutation would affect the interaction partners and the production of force within the sarcomere. The force production, in turn, affects the entire cardiomyocyte mechanically or by accumulation of metabolites and other changes in the intracellular milieu. Finally, on a higher hierarchical level, both the total force and differences in force production between neighboring cells affect the entire organ and its function. This process is modulated non-linearly by non-genetic factors and various cellular adaptive responses, e.g., activation of a range of cell signaling pathways (see below) and various posttranslational modifications. Not the least, this includes phosphorylation of a range of sarcomere proteins [64,65,66], in particular, RLC [28,67,68], troponin [69,70] and cMyBP-C [71,72,73]. Overall, the tissue response to cope with the changes caused by a sarcomere mutation over time alters the tissue morphology and function by cardiac remodeling, e.g., hypertrophy in HCM (see further Introduction).

### 2.3. Diastolic Dysfunction Progressing to Diastolic Heart Failure

During the diastolic phase of the cardiac cycle, the ventricles relax after the preceding systolic contraction to allow filling of the ventricles with blood before the next systole. Due to a variety of medical conditions, the ventricles can become “stiff” and/or unable to fully relax. This compromises ventricular filling, making the ventricles pump smaller quantities of blood during each heartbeat. As outlined below, diastolic dysfunction is often seen in HCM. Over time, diastolic dysfunction progresses into diastolic heart failure as blood accumulates in other parts of the body (mainly lungs or legs).

### 2.4. Calcium Sensitivity

Before force- and motion-generating cross-bridge cycles can take place between myosin motor domains and actin filaments, the activation signal needs to reach the muscle and increase the cytosolic concentration of calcium ions. The calcium entering the cardiomyocyte from the extracellular fluid, but particularly that being released to the cytoplasm from the sarcoplasmic reticulum, binds to the troponin complex on the thin filaments to initiate contraction (Figure 2). The Ca^2+^-concentration at which force is half-maximum (typically ~1 µM) is a measure of the calcium sensitivity (Ca-sensitivity below). However, calcium also (Figure 2) activates critical cell-signaling pathways [74] and enzymatic processes that modulate contractile function [75]. Finally, Ca-overload, as well as increased myofilament Ca-sensitivity [76], has arrythmogenic and other deleterious effects. The binding of calcium to troponin is the central initiating event in contractile activation. Thus, upon such binding, the connected tropomyosin molecules move on the thin filaments to uncover the myosin binding sites on actin. The Ca-sensitivity is determined by the Ca-affinity of troponin C, as well as by other properties of both the thin filaments (e.g., flexibility of tropomyosin) and the thick filaments (e.g., RLC and cMyBP-C phosphorylation and tension in the filament; see Reference [77]). In failing hearts, the calcium sensitivity can either increase or decrease depending on the specific disease. An increase in calcium sensitivity is expected to be associated with increased contractility, as well as impaired relaxation [78].

### 2.5. Tension Cost and Energy Efficiency

During the cross-bridge cycle, muscles convert free energy from ATP turnover into mechanical work with tension generation and shortening of the cardiomyocytes. In papers on HCM, the energy efficiency is often reported as the ratio between work and energy consumption during the cardiac cycle, where energy usage is measured directly as ATP consumption or indirectly from O_2_ consumption [80]. An alternative estimate of the contraction efficiency is the “tension cost”, which is used in some studies. This is defined as the ratio between the developed tension (usually isometric; per muscle cross-section) and the energy required to produce that tension level [21,81].

### 2.6. Interacting Head Motif (IHM), Super-Relaxed (SRX) and Disordered Relaxed (DRX) States of Myosin

The interacting head motif (IHM) [82,83] refers to a configuration with interaction of the myosin heads with each other while also folding back onto their coiled-coil tail with sequestration (“parking”) of the heads on the thick filament backbone. The ATP turnover and actin binding are both markedly inhibited in this configuration, where the heads presumably also interact with cMyBP-C and titin. The very low ATP turnover of the sequestered heads led to the concept of a super-relaxed state (SRX) [84,85]. Myosin heads in this state display ~10 times lower basal ATP rate (~0.003 s^−1^) than isolated myosin heads (subfragment 1) free in solution (~0.03 s^−1^). The latter have properties similar to those of myosin molecules in the thick filaments in the disordered relaxed state (DRX) [84], where myosin heads protrude toward the actin filament without actually binding. There is evidence that several factors operate to shift the population from the SRX to the DRX state in a normal heart upon transition from relaxation to active contraction. This includes phosphorylation of the regulatory light chains of myosin and the cMyBP-C, as well as increased tension in the thick filaments of the sarcomere [28,67,68,71,72,73,77,86,87,88].

As considered further below, there is growing conviction in the field that many, if not all, HCM mutations cause the deleterious remodeling of the heart by producing hypercontractility, primarily as a consequence of increased population of the DRX state under various conditions [15,26,89]. Thus, mutations in the flat mesa region of the myosin heavy chain, as well as in the converter domain and the S2-region, reduce the formation of the IHM [26,89,90], as well as mutations in cMyBP-C [91] and the myosin RLC [92]. If the SRX population is thus reduced, the force production and calcium sensitivity increase, as well as energy usage, particularly at low calcium-activation levels.

## 3. Different Disease Models

The different model systems for studies of HCM, range from isolated proteins to whole organs, using samples either from humans or experimental animals, each approach with its specific advantages and challenges. Different model systems have been considered and compared briefly in recent reviews [24,41,45], one with specific emphasis on the disease models per se [47]. Here, we briefly review key disease models as an important background to the experimental evidence for and against the three major hypotheses (Table 1). Our focus is on results derived from preparations of human origin, and, compared to a recent methodological review [47], we put more emphasis on studies using isolated proteins.

### 3.1. Human Disease Models

#### 3.1.1. Isolated Proteins

Human sarcomere proteins with HCM mutations can either be isolated from cardiac (sometimes skeletal muscle; see References [28,55,93]) tissue from patients or be expressed in cell systems, such as *Escherichia coli* (TnI, TnT, Tm, cMyBP-C and myosin light chains) [94], insect cells (actin; see References [95,96]) or C2C12 myotubes (myosin heavy chains; see References [97,98,99]) (Figure 3). The isolated proteins can then be subjected to a range of investigations, e.g., structural studies (X-ray crystallography [26,100], cryo-EM [101] and high-speed AFM [102]), to elucidate structural effects of the mutations, and transient and steady-state biochemical kinetics [99,103,104] to elucidate changes in the ATP turnover mechanism. For insight into effects of mutations on mechanical and elastic properties of relevance in cardiac contraction, “unloaded” [105,106] and “loaded” [107] in vitro motility assays are often performed [99,108,109,110,111]. In these assays, fluorescence labeled actin filaments or thin filaments (e.g., with mutations in thin filament proteins) are observed as they are propelled by myosin motor fragments immobilized on suitably modified glass surfaces. Additionally, optical tweezers and cantilever-based force measurements can be used to directly measure the forces developed by single myosin motors or small motor ensembles when they interact with an actin filament or a thin filament (actin + regulatory proteins) [99,112]. Moreover, myosin and actomyosin ATP turnover kinetics can be probed by single-molecule ATPase assays [113,114], using total internal reflection fluorescence (TIRF) microscopy and fluorescent ATP.

Proteins, both with and without mutations, can be isolated from hearts of patients with HCM and donor hearts, respectively. The patient material is obtained from tissue removed during surgery, e.g., to eliminate left ventricular outflow tract obstruction or from heart transplantations, often from special facilities, e.g., The Sydney Heart Bank [115]. However, in the case of β-MHC mutations, proteins can also be isolated from biopsies taken from slow skeletal muscle expressing the same myosin heavy chain isoform as the cardiac ventricles [93,111,116].

One major advantage of using proteins isolated from patients is the relevance from a human disease perspective. However, there are also challenges in using such proteins, in addition to the difficulties of obtaining significant amounts. First, the method of obtaining tissue directly from human hearts necessarily produces a bias toward patients with severe disease, and it is virtually (e.g., from an ethical perspective) impossible to obtain human heart material from mutation carriers without hypertrophy or other overt signs of pathologic remodeling. In this context, it is, however, of interest to note that recent studies of β-MHC mutations in a transgenic rabbit model seemed to give similar results whether preparations were derived from cardiac ventricles or slow skeletal muscle (*musculus soleus*) [117]. Myosin from skeletal muscle is, of course, appreciably easier to obtain by biopsies, with less severe ethical considerations.

Other challenges in using patient material relate to the fact that HCM is generally an autosomal dominant disorder and the patients are mostly heterozygous with one allele coding for wild-type protein and the other for mutated protein. Furthermore, the fractional expression of wild-type and mutated protein may vary, not only between patients and between the specific mutations but also between cells in a given patient [28,33,34,40,118,119]. Finally, the proteins isolated from patients are likely to be affected by different types of posttranslational modifications due to the (secondary) remodeling process. All of this gives an inhomogeneous protein preparation, with a varying and poorly characterized mixture of wild-type and mutated proteins and with a variable degree of posttranslational modifications.

The alternative approach to obtain human protein, whether wild-type or mutated, is to use cell based expression systems. This allows for the production of well-characterized homogenous protein preparations with regard to amino acid sequence; that is, all proteins are either wild-type or have a predetermined mutation. Furthermore, the proteins have not been subjected to a pathologic remodeling process as in the case with proteins isolated from patients. This follows because the proteins are produced in cell systems without contractile effects of the mutations believed to be of crucial importance in triggering the remodeling. However, there are also challenges with the use of such expressed proteins. It is important to ensure that the proteins both fold appropriately and are equipped with posttranslational modifications essential for normal function (see also Reference [120]). For most of the sarcomere proteins (see above), expression in *E. coli* allows proper folding and function. For other proteins, such as actin and myosin, expression in *E. coli* is not possible due to lack of necessary chaperones to assist proper folding. Cardiac (and skeletal muscle) myosin heavy chain is particularly laborious to obtain in a fully folded functional form, as it requires specific chaperones that are only expressed in mammalian striated muscle cells, differentiated to the myotube level [98,121]. Therefore, recombinant human striated muscle myosin isoforms are exclusively overexpressed and purified from C2C12 myotubes (Figure 3), a technique that is in routine use only in a limited number of laboratories.

The studies of expressed proteins, e.g., in the in vitro motility assay or biochemical kinetics experiments, is ideally performed with all interaction partners (actin, myosin, regulatory proteins, etc.) from human cardiac ventricular muscle. However, in practice, hybrid systems are often used where the protein with the mutation is of human origin (e.g., expressed from the human gene) but other proteins have other origins. For instance, some studies of mutations in cardiac myosin, using the in vitro motility assay, are performed by using actin from skeletal muscle [99]. This is probably not a concern, because the skeletal and cardiac muscle actin differ only by four amino acids with conservative substitutions. Moreover, no differences in velocity were found between wild-type cardiac and skeletal muscle actin propelled by wild-type motor fragments from fast skeletal muscle myosin [122]. Hybrid systems are also used for investigating effects of mutations in other than the *MYH7* gene. Thus, frequently, human Tm, TnI and TnT (wild type or with mutations) expressed in *E. coli* are used to reconstitute skeletal muscle actin from rabbit into thin filaments, followed by testing in the in vitro motility assay, where they are propelled by fast skeletal muscle myosin II. This raises some concern because the fast skeletal muscle myosin is appreciably faster and differs with respect to several kinetic parameters, due to differences in both the myosin heavy chain and light chain isoforms (cf. Reference [123]). However, this concern is dampened by recent in vitro motility assay studies [124], using only cardiac proteins. Importantly, these studies suggest similar effects of two different mutations in Tm on both calcium sensitivity and maximum sliding velocity, as previously observed by using in vitro motility assays with skeletal muscle myosin and actin [125,126].

In addition to the complexities considered above, one may question to what extent studies of isolated proteins that use very few of the sarcomere components give results that are relevant for understanding human disease. This issue is exemplified by studies of functional effects in isolated protein ensembles due to the R403Q β-MHC mutation. These effects differed [127] if regulated thin filaments (actin + troponin + tropomyosin), rather than isolated naked actin filaments, were used for in vitro motility assays, single molecule force measurements and studies of actin activated ATPase. For some other β-MHC mutations, however, no differences were observed between thin filaments and isolated actin filaments [18].

When using isolated proteins, it is also important to consider whether the full-length protein or various protein fragments are used, e.g., for MyBPC and myosin. With regard to myosin, different fragments of the entire molecule, rather than the full-length heterohexamer (Figure 1), are often used for biochemical (e.g., transient kinetics) and biophysical studies (e.g., force measurements) of either single molecules or protein ensembles. Motor fragments (Figure 1C) that are often used include subfragment 1 (S1) and heavy meromyosin (HMM). Subfragment 1 consists of the myosin motor domain, including the essential light chain (ELC; short S1) or both the essential and regulatory light chain (RLC; long S1). HMM, on the other hand, consists of both heads of myosin (including both the RLC and ELC of each head) and the subfragment 2 part of the myosin tail. Due to challenges to express and purify sufficient amounts of (cardiac) full-length HMM fragments, the constructs used generally have shortened subfragment 2 parts [89]. This difference between the fragments is essential to consider when selecting a preparation for studies of a certain phenomenon. Thus, clearly, any effects of mutations in the RLC can only be studied by using the HMM or the long S1 fragments because the short S1 fragment does not include the binding site for the RLC. Moreover, the effects of mutations on interactions between the two myosin heads or between the heads and the tail require the use of full-length myosin or two-headed myosin motor fragments, such as HMM or similar. Of particular interest in this connection is the role of the interacting head motif (IHM) [83] wherein the two heads interact with the S2 part of the myosin tail and with each other to form the super-relaxed state (see above). This effect seems to be further enhanced when myosin molecules are incorporated into filaments [128].

#### 3.1.2. Myofibrils and Skinned Muscle Cells

Skinned muscle cells, or rather skinned trabeculae with arrays of coupled ventricular myocytes, can be isolated from heart tissue. These preparations contain cells whose membranes have been permeabilized by chemical treatment while maintaining the myofibrillar system intact. This allows studies of the contractile machinery at predetermined concentrations of calcium, ATP, etc., added to the solution that surrounds the cells. Studies of force generation, shortening and other mechanical and kinetic properties can also be performed by using the thread-like, approximately 1 µm–wide myofibrils that fill the myocardial cells [117,129,130]. The myofibrils and cells isolated from human cardiac tissue are subject to similar advantages and challenges, as described above for isolated proteins. Thus, one important advantage is the clinical relevance, whereas challenges include effects of different degrees of remodeling and heterogeneities in expression levels and posttranslational modifications.

A major advantage of myofibrils and isolated skinned muscle cells compared to isolated proteins is that functional effects of the mutations can be studied under intact 3D order of the sarcomeres where all protein interaction partners are of human cardiac origin. However, the complexity, as well as the risk for secondary effects due to remodeling, is higher than for the isolated proteins. The remodeling effects are, of course, not an issue if the aim is to characterize details of the fully developed phenotype, but they become problematic if the desire is to elucidate the primary mutation-induced changes that initiate the remodeling. In this regard, isolated myofibrils have advantages over cells by circumventing some effects of remodeling, such as lower myofibrillar density and myofibrillar disarray in HCM. Mechanical studies of myofibrils also facilitate the elucidation of transient mechanical changes upon solution exchange due to very short diffusion distances. Such changes are not so readily studied by using skinned muscle preparations. The myofibrils, on the other hand, are more fragile than the cellular preparations and require more advanced equipment for force measurements (cf. References [15,26,89]). As is the case with isolated proteins, myofibrillar and skinned muscle cell preparations for studies of HCM-causing mutations in the β-MHC can also be obtained from slow skeletal muscle that expresses β-MHC in high quantities [28,93].

#### 3.1.3. Intact Muscle Strips from Living Hearts

Compared to skinned muscle cells, preparations with an intact membrane (living trabeculae or papillary muscles) have the advantage that they more faithfully mimic the normal cardiac contraction. Thus, by electric stimulation, the physiological Ca-transient can be reproduced, together with the associated twitch contraction that takes place during a normal heart beat. Such physiologically faithful transient responses have important advantages compared to contractions at different constant Ca^2+^ concentrations typically applied in skinned muscle cells and myofibrils when the Ca^2+^ sensitivity of contraction is assessed. However, it would seem extremely challenging to obtain intact muscle strips from human cardiac tissue. In contrast to the case with skinned cells and myofibrils, the tissue cannot be frozen before isolation of living cells, and, to the best of our knowledge, no studies of intact human cardiac muscle strips have been reported. However, see below for the use of animal models.

#### 3.1.4. Engineered Human Cells and Tissues

In recent years, appreciable efforts have been made to circumvent the challenges of obtaining living cardiac cells from human hearts by engineering such cells (and tissues) using stem-cell technology, more specifically human induced pluripotent stem cells (hiPSC; cf. review in Reference [131]). In this process, human fibroblasts or other somatic cells are first reprogrammed to the immature pluripotent stem-cell stage, followed by targeted differentiation into human cardiomyocytes [32]. Such engineered myocytes can usually be electrically excited to produce an intracellular Ca^2+^ transient and a twitch-like contraction in similarity to mature living ventricular cells. The cells also exhibit a protein expression pattern partially similar to adult cardiac myocytes (however, see References [47,132]). Furthermore, they are filled with myofibrils showing a regular cross-striated sarcomere pattern, albeit exhibiting somewhat different overall morphology than adult myocytes (Figure 4A,B). The cells can be further engineered, together with cells of other types, to form sheets of tissue useful for mechanical measurements, e.g., forces and displacements (Figure 4C). A major advantage of these engineered cells and tissues is that they can be obtained in large abundance. Moreover, in addition to their similarities to adult human cardiomyocytes, they can be genetically engineered, for example, using CRISPR/Cas9 methodology. This allows for the specific and faithful generation of HCM mutations as desired in a well-controlled genetic background [47].

Whereas the engineered cardiac cells have appreciable similarities to fully differentiated adult cardiomyocytes, the correspondence is not perfect with lower maturity and some structural and gene-expression differences [47]. Possibly, the differences relate to the lack of influence of mechanical forces and a 3D scaffold of other cells that are likely to be important for the normal development of cardiomyocytes in the human heart. Nevertheless, comparison with other experimental systems (e.g., Reference [27]) suggest that engineered cells based on the hiPSC technology constitute a valuable addition to the arsenal of disease models for the study of HCM. For more details on this approach, as well as a discussion of possible pitfalls, we refer to recent reviews by others [24,47,49].

It has been suggested that cardiomyocytes derived from hiPSCs demonstrate primary mutation effects [47]. However, hiPSC derived cardiomyocytes with HCM mutations often show significant hypertrophy of individual cells [27,32,49,133,134], generally seen as a hallmark of the fully developed HCM phenotype, i.e., a consequence of long-term remodeling. Additionally, myofibrillar disarray has been demonstrated in the differentiated cells derived from hiPSC (cf. References [32,49,133]), something that is also generally associated with secondary remodeling. However, evidence from other types of studies supports the idea that myofibrillar disarray may occur early in the pathogenesis. This includes studies of mouse embryo hearts [135] and findings that the β-MHC mutations R403Q, R453C and G548R cause myofibrillar disarray when incorporated into myofibrils of chicken embryonic cardiomyocytes in culture [63]. The importance of mechanical forces on cellular development and myofibrillogenesis in these cases is suggested by findings that myosin-stimulating compounds applied during development affect the phenotype [136]. Either way, it is important to clarify the full meaning of these findings before using engineered human cells as model systems for initiation of the pathologic remodeling in HCM.

#### 3.1.5. Whole Hearts and Imaging

In addition to gross morphologic and histologic examination of whole hearts obtained postmortem or in heart transplantations, various forms of cardiac imaging provide rich information. Such approaches include different versions of echocardiography and magnetic resonance imaging (cardiac MRI). In addition to providing pure morphologic information, it is possible to assess functional parameters related to diastolic and systolic function, non-uniformities of cardiac strain and pressure gradients, as well as alterations in cardiac metabolism. For the reader interested in more information on this exciting and expanding field, we refer to recent reviews [137,138,139]. However, we also discuss key studies from recent work using cardiac imaging below.

### 3.2. Transgenic Animals

Some results from transgenic mouse models of HCM seem to be quite similar to those derived by using human disease models [27]. However, other studies suggest that the different physiology of mice and humans, e.g., ~10-fold higher heart rate in mouse, and the associated expression of fast protein isoforms may produce different effects of a mutation in mouse compared to the corresponding mutation in humans (cf. References [140,141]).

For an overview of results obtained by using transgenic mice, as well as the pros and cons of transgenic mouse models, we refer to reviews by others [24,41,45]. However, we would like to stress some advantages of the animal models. For instance, it is feasible to study preparations on all hierarchical levels (from isolated proteins to living cells and whole hearts) and hearts on different stages of development before and after extensive remodeling. This, together with the short life span of mice, makes it possible to follow the natural course of the disease much more readily than in humans.

One example of a study illustrating the mentioned benefits is that by Song et al. [142], which used mouse ventricular preparations to investigate effects of a mutation in actin (*ACTC1* E99K). Most notably, it was found that the isometric force development of isolated myofibrils at full Ca^2+^-activation was unchanged compared to the wild type in the presence of the mutation. In contrast, twitch forces of intact papillary muscles were 3-to-4 times higher than seen in the absence of the mutation. This was attributed to the increased calcium sensitivity produced by the mutation.

Another example demonstrating key benefits of the transgenic mouse model is a recent study [143] showing that the drug mavacamten inhibits the HCM pathogenesis in mice. When chronically administrated from young age, the drug inhibited the development of ventricular hypertrophy, cardiomyocyte disarray and myocardial fibrosis in mice with heterozygous human HCM mutations in the myosin heavy chain gene. This study based its conclusions on experiments using preparations on different hierarchical levels from isolated molecules to whole hearts and on distinct stages of the pathogenesis from young mice without the characteristic HCM phenotype to adult mice with fully developed HCM phenotype in the absence of the drug.

Finally, studies of homozygous cMyBP-C knockout (cMyBP-C −/−) mice support ideas on how mutations in this protein might lead to HCM (see also Reference [15]). First, the idea that cMyBP-C mutations cause HCM by lack of the protein through haploinsufficiency is supported by findings of severe cardiac ventricular hypertrophy of cMyBP-C −/− mice associated with myocyte disarray and fibrosis [144,145,146]. Second, studies of isolated muscle fibers provide evidence for hypercontractility with increased cross-bridge cycling rates and increased power output of muscles from such mice [147,148]. Finally, structural studies revealed more disordered myosin heads in muscles of cMyBP-C −/− mice compared to WT, with the heads having moved away from the thick filament surface [149]. This fits with an idea that the deficiencies (or loss) of cMyBP-C in HCM cause hypercontractility by destabilizing the IHM of the myosin heads on the thick filament backbone.

## 4. Three Major Hypotheses for the HCM Pathogenesis

Below we consider three hypotheses for the pathogenesis of HCM (Table 1), assuming that they operate in all genetic forms of HCM, due to mutations in sarcomere proteins. As already mentioned above, we are aware that this view may be too simplistic [24,41,42]. However, we believe that the general hypotheses capture important features of the pathogenesis despite possibly being modulated by specificities related to each individual mutation. Alternatively, the hypotheses may account well for a large fraction of the HCM cases, but with some exceptions that do not fit into the general picture.

### 4.1. Literature Search

For the hypercontractility hypothesis, the most detailed analysis was performed for papers from 2011 to 2021 that were found by using PubMed (20 November 2021) and the search terms (hypercontractility OR hypercontractile) AND (HCM OR “hypertrophic cardiomyopathy”), resulting in 60 hits, out of which 32 were published in 2019–2021. All of these papers were considered in full, except for case reports. In addition, we considered papers that we found based on the search terms (relaxation OR “diastolic dysfunction”) AND (HCM OR “hypertrophic cardiomyopathy”). In this case, we obtained 458 hits in total (from 2011), out of which 150 were published in 2019–2021. Here, we only considered papers in full based on an initial selection from the titles. Finally, we also read several recent review papers [14,15,21,24,39,41,45,46,47,48,49] and selected additional original papers from these.

The search strategy for the literature related to the hypocontractility hypothesis was different from for hypercontractility, particularly because hypocontractility has been considered to a limited degree in recent years. We therefore performed searches without time limitations. The search queries used on PUBMED and SCIENCE DIRECT (end of February 2021) were “hypertrophic cardiomyopathy AND (hypocontractility OR hypocontractile) AND myosin”, resulting in 14 and 71 hits, respectively. We also considered any relevant hit found on the first search pages, using Google Scholar and Semantic Scholar, an AI powered research tool for the scientific literature. Out of the papers found in these searches, we included only papers with results obtained strictly using human samples. Finally, by reading selected papers and certain reviews [11,47,48,150], we selected additional papers from these to further elucidate the hypocontractility hypothesis.

The searches for the non-uniformity hypothesis were performed with a wider scope, using PubMed Central on 1 September 2021. The search phrases (“non-uniformities” OR “contraction imbalance”) AND (HCM OR “hypertrophic cardiomyopathy”) were first used for a search that gave 26 hits, out of which six were published in 2019–2021. Due to a limited number of relevant papers found, the search was changed to use the search terms (“homozygote” OR “homozygous”) AND (HCM OR “hypertrophic cardiomyopathy”) with 4129 hits, with 1243 published in 2019–2021. An initial selection on titles gave a selection of papers that we considered in full. Furthermore, several original papers were chosen from more general review papers [24,57,151,152,153] and from one of the papers [36] where the hypothesis was initially explicitly stated.

### 4.2. Hypothesis 1: Hypercontractility as the Primary Disturbance

The hypercontractility hypothesis in the form evaluated here states, hypercontractility, defined as in the Introduction (Table 1), is the primary mutation-induced characteristic that initiates pathologic remodeling in HCM.

#### Origin and Development of the Hypercontractility Hypothesis

While first recognizing that the term “hypercontractility” might include a range of phenomena and could have different meanings to different researchers, we believe that the hypercontractility hypothesis can be traced back to a clinical imaging study showing evidence for diastolic dysfunction in HCM mutation carriers [25] and a review paper by Ashafrian et al. [11]. The latter paper summarized some experimental findings that were available at the time showing that HCM mutations may cause either hypo- or hyper-contractility on the molecular and cellular levels. Additionally, Ashafrian et al. [11] cited the just mentioned clinical study [25] as lending further support to the importance of hypercontractility in HCM pathogenesis. This seemed to argue against the dominant hypothesis at the time (Figure 5A), that hypocontractility is the primary change in HCM leading to compensatory hypertrophy. However, while bringing hypercontractility on the table, Ashafrian et al. [11] recognized the inconsistencies in the available experimental results. They therefore moved on to postulate that inefficient ATP usage and/or energy depletion is the main common denominator that triggers pathologic remodeling in HCM (Figure 5B). Evidence for inefficient ATP usage was later corroborated in experimental systems on different hierarchical levels from isolated proteins to whole hearts [21,23,27,81,133] (see further below). Additional support for a central role of energy depletion in triggering pathologic HCM remodeling comes from the findings that not only sarcomere protein mutations, but also mutations of key proteins in metabolic pathways, lead to a cardiac phenotype of the HCM type [11,13]. Furthermore, disturbed metabolic signaling and mitochondrial dysfunction seem to be quite central, more generally, in HCM [154]. Here, we do not treat inefficient energy usage or energy depletion as primary effects of the HCM mutations in sarcomere proteins, and the phenomenon is not included among our three key hypotheses (Table 1). Instead, we view these effects as downstream consequences of either hypercontractility, hypocontractility or contractile non-uniformities.

The hypercontractility hypothesis (Figure 5C) has been elaborated upon in greater detail recently. Particularly, stimulated by studies of isolated proteins such as human beta-myosin expressed in C2C12 cells [99,155,156], Spudich [14,16] proposed that a key facet of hypercontractility is increased power production on the myofilament level. However, whereas several studies of myosin subfragment 1 (S1) suggest that hypercontractility is of importance, a clear-cut association between HCM severity and hypercontractility could not be consistently found in subsequent studies using isolated myosin S1 [18,104].

Power output (=force × velocity; F × v), is determined by the load-dependent kinetics of the actin–myosin interaction that underlies the force–velocity relationship. At a fixed velocity, the power output is proportional to the ensemble force, F_ens_, attributed to the huge number of myosin motors that currently interact with all thin filaments in the cardiac sarcomeres. This force is by a (highly) simplifying approximation, given [15] by the following relationship:F_ens_ = F_int_·N_a_·t_s_/t_c_(1)
where F_int_ is the intrinsic force of the myosin molecule (e.g., as measured from single molecules, using optical tweezers) and N_a_ is the number of functionally accessible heads for interaction with actin. Furthermore, t_s_/t_c_ is the duty ratio that is determined by the kinetic parameters of the actin-activated ATPase cycle of myosin at the given velocity, i.e., the ratio between the time myosin spends in a state strongly bound to actin (t_s_) and the total ATPase cycle time (t_c_). Whereas changes in these parameters, caused by HCM mutations in β-MHC, in many cases, would lead to increased ensemble force, the association is not clear-cut [15]. This leaves changes in the number of available motors N_a_ as a remaining parameter to consider. This number is usually only a rather small fraction of the total number of myosin motors in the thick filaments [16]. One reason is a rather low level of activation of the thin filaments related to binding of Ca^2+^ to a low fraction of the regulatory protein complex of troponin–tropomyosin, thereby only providing access to a similarly limited number of myosin binding sites on actin. Clearly, increased Ca^2+^ sensitivity of the thin filament regulatory system, due to HCM mutations (e.g., in TnT, TnI, Tm or β-myosin), could increase F_ens_ by increasing N_a_. However, one further factor could work in similar direction, namely a shift of myosin motors on the thick filament backbone from the SRX to the DRX state. In addition to increasing N_a_ during active contraction, this effect might also increase the ATP consumption at rest because DRX myosin heads have approximately 10-fold higher basal ATPase rate (without actin-binding) than SRX-heads. The idea that destabilization of the IHM, with increase of the DRX/SRX ratio, is important was proposed by Spudich and co-workers [15,89,157], partly prompted by the findings of unclear association between HCM severity and hypercontractility, as deduced from changes in F_int_, t_c_ and t_s_ in studies using S1 motor fragments. The idea was also based on the finding that β-MHC mutations accumulate in a flat region on the myosin molecule, the myosin mesa [90]. The mutations in this region were predicted to destabilize the IHM, thus increasing the availability of myosin motors for interactions with the thin filaments. More recently, quite extensive evidence has accumulated (see below) to support the importance of the destabilized IHM as a factor underlying hypercontractility due to both β-MHC and cMyBP-C mutations. The hypercontractility hypothesis is also consistent with results from a study [158] that identifies a relationship between the integral of the myofilament tension developed over time and heart growth where an increased tension integral promotes concentric hypertrophy as in HCM.

In summary, key characteristics that we include under the heading of hypercontractility include (i) increased ATP turnover and cross-bridge cycling rates, (ii) increased force, shortening/sliding velocity and power output (force x velocity) and (iii) diastolic dysfunction on the whole-heart level related to increased Ca-sensitivity and/or destabilization of the IHM with slowed relaxation. Each of these aspects of hypercontractility could lead to energy depletion as a secondary effect, associated with accumulation of various metabolites with subsequent effects on cell-signaling systems.

### 4.3. Hypothesis 2: Hypocontractility as the Primary Disturbance

Despite the currently predominant view that the primary disturbance of cardiac β MHC mutations is hypercontractility (see above), we here attempt to defend the opposite idea that hypocontractility is the primary mutation-induced disturbance that initiates remodeling to the HCM phenotype. That mutations will cause a weaker heart is intuitively appealing. Considering an otherwise extraordinary biological pump that is powered by defect motors, it is not far-fetched to conclude that the pump will also be defect and contract ineffectively. Accordingly, the idea of heart hypertrophy in HCM as a compensatory mechanism to counteract poor motor function was one of the first “unifying” hypotheses proposed [150,159,160]. Despite being later criticized as inconsistent with laboratory findings and clinical evidence [11], recent results suggest that it may be worth to revisit the hypothesis, at least for explaining a portion of the HCM cases. For instance, a systematical review of data from human skinned muscle strips and myofibrils obtained from frozen heart samples with HCM mutations in different sarcomeric genes revealed that the maximal developed force averaged ~40% lower than in control samples [48]. Although this only applies to conditions of full activation that are not seen during physiological cardiac contractions, it is nevertheless quite striking and worth to consider further. Other results of interest in this connection include some findings from cardiomyocytes and heart tissues engineered from hiPSCs [32]. Finally, some echocardiography-based strain-imaging studies have reported evidence suggesting hypocontractile function in HCM patients [161].

The term “hypocontractility”, as we use it, may be defined as the opposite condition of hypercontractility (Table 1). Thus, in addition to lower contractility, as the name suggests, with lower force production in the living heart, as indicated by cardiac imaging, we have included several measurable outcomes on different hierarchical levels. These include (i) lower basal and actin-activated ATP turnover (reduced k_cat_), (ii) lower (loaded and unloaded) velocity of actin filaments in the in vitro motility assay, (iii) lower population of the force holding actomyosin ADP state (resulting in lower duty ratio) and (v) lower calcium sensitivity. In Table 2 and Section 4.4, we consider evidence for hypocontractility and hypercontractility side by side with the data for hypocontractility, primarily focusing on β-MHC mutations.

### 4.4. Different Perspectives Related to Hypo- and Hypercontractility Hypotheses from Experiments on Different Hierarchical Levels

#### 4.4.1. Studies on Isolated Proteins

With β-MHC mutations, it has been found in several studies using isolated S1 motor fragments (without possibility to form the IHM) that mutations causing hypercontractility often are associated with HCM, whereas mutations causing hypocontractility often are associated with dilated cardiomyopathy (DCM) [112,174]. In most of these studies, the S1 motor fragments of β-MHC have been obtained by using the current golden-standard expression method based on C2C12 mouse myotubes [98,175,176] (Figure 3) to obtain homogenous preparation of human muscle myosin constructs. However, despite the increased tendency for hypercontractility in HCM, there are notable exceptions. Thus, using S1, only minimal or no hypercontractility was produced by HCM, causing mutations in the converter domain of myosin (R719W, R723G and G741R) [18]. For the R719W and R723G mutations, intrinsic force was lower albeit accompanied by faster gliding velocities of actin filaments, as well as regulated thin filaments. In another study, using S1 constructs, particularly the P710R mutation showed signs of hypocontractility with large reduction in k_cat_ and the motor duty ratio [104]. Moreover, the apparently quite severe [177], but rare, R712L mutation [17] led to severe hypocontractility on the molecular level, with an inhibited working stroke as suggested by studies using expressed HMM constructs from beta-cardiac myosin. Similar findings were obtained recently for some mutations (D778V, L781P and S782N) in the pliant region of the myosin lever arm [29].

Limitations of studies using the S1-like constructs is that any reported results could be effectively overridden if the motor is examined in the natural sarcomeric organization particular by the effect of interacting heads, e.g., the IHM. Thus, two-headed HMM-like constructs bearing HCM mutations are more desirable to study in order to examine DRX/SRX effects where the gain-of-function (hypercontractility) may be revealed despite myosin exhibiting hypocontractility as a one-headed construct. It has thus been found [178] that both the abovementioned R719W [18] and the P710R [179] mutations may cause hypercontractility by destabilizing the IHM (see further below). However, such a difference between S1 and two-headed myosin motor fragments does not seem to exist with the mentioned mutations in the pliant region of the lever arm [29]. Moreover, it is unknown whether the severe molecular hypocontractility, as seen with the R712L mutation, might be transformed into hypercontractility, either by reduced tendency for formation of the IHM or changes in the Ca-sensitivity by other mechanisms. Finally, to expand this picture further, there was no clear correlation between HCM severity and the level of hypercontractility in studies of several β-MHC mutations, using expressed myosin subfragment 1 [104]. In some cases, however, severe forms of hypercontractility were observed [156] with the severe early onset disease mutations D239N and H251N on the myosin mesa region. This included increased actin activated ATPase activity, increased sliding velocity in the in vitro motility assay and increased intrinsic force. Adhikari et al. [156] also found reduced interaction of the S1 domain with myosin S2 fragments in solution for these mutations suggesting the possibility of a destabilized IHM. An increase in sliding velocity is in line with an increased cross-bridge detachment rate at full activation [180,181,182], as suggested to occur with several mutations in myofibril studies (see further below). On the other hand, full-length β-myosin, purified from slow skeletal muscle biopsies from different HCM patients, allowed examination of seven different mutations (T124I, Y162C, G256Q, R403Q, V606M, R870H and L908V) by in vitro motility assays showing consistently lower actin filament velocities compared to healthy controls [111,116]. Similar findings were made in later studies, again with β-myosin isolated from biopsies of soleus muscles in patients with the R403Q mutation in the β-MHC [162]. In the latter case, both the maximum sliding velocity in the in vitro motility assay and k_cat_ of the actin-activated ATP turnover rate were reduced. In yet another study [183], however, faster actin filament velocities were noted for the myosins carrying mutations R403Q and L908V isolated from the same patient biopsies as in the earlier work of Cuda and co-workers [111]. Results in accordance with the latter view were obtained by using myosin purified from a biopsy from a rare patient being homozygote for the R403W β-MHC mutation [184]. This myosin exhibited increased sliding velocity of actin filaments in the in vitro motility assay, as well as increased k_cat_ of the actomyosin ATPase and much weakened apparent affinity for actin (K_app_) compared to the control. As the authors suggested, this implies inefficient ATP utilization and reduced mechanical efficiency, e.g., as consequence of reduced duty ratio (lower population of force bearing states) due to weakened actomyosin interactions.

Finally, with regard to studies of isolated β-myosin in HCM, the pioneering β-myosin protein expression work of Sata and Ikebe [185] deserves to be mentioned. These authors expressed and purified β-myosin constructs (amino acids 1-1138), carrying R249Q, R403Q, R453C and V606M HCM mutations. The results of this study may be criticized due to use of an insect-cell-based expression system for the β-MHC in contrast to the contemporary golden standard, using mouse C2C12 muscle cells. Nevertheless, the relative comparison between the wild-type and mutated motor may still be valid. This comparison showed that all the studied mutations (expect V606M) reduced k_cat_ and actin filament motility where the mutation R403Q showed the greatest effect, also greatly weakening the apparent dissociation constant for actin (K_app_).

For thin filament mutations, there does not seem to be generally consistent effects on power, force and velocity under full calcium activation. However, increased calcium sensitivity has been observed as a result of HCM causing mutations in tropomyosin or troponin T, based on ATPase assays [155] and in vitro motility assays [109,124,125,126,169,186,187,188,189], using regulated thin filaments. Another effect that has often been observed in the in vitro motility assay, using troponin from HCM patients, is the lack of effects of TnI phosphorylation on Ca^2+^-sensitivity (uncoupling). In these in vitro motility assay studies, the effects seem to be associated with apparent (not fully understood) [170] changes in troponin T of the HCM patient, whether the mutations are in the thin filaments or other sarcomere proteins (e.g., β-MHC and cMyBP-C).

Due to the complexities and only partly understood roles of cMyBP-C in coordinating thin and thick filament based regulation [15,73], studies of isolated proteins can only tell part of the story. Therefore, despite the high relative frequency of cMyBP-C mutations in HCM, we do not consider studies of isolated cMyBP-C in detail. However, it deserves to be mentioned that cMYBP-C mutations have been postulated to cause hypercontractility in HCM by effects both on the thin and thick filaments [190], in the latter case, potentially related to destabilization of the IHM of myosin [15]. See Section 3.2 for related discussions

As demonstrated in the paragraphs above, the hypercontractility hypothesis may often be rescued, despite conflicting results from studies of myosin S1, if the HCM causing mutations destabilize the IHM. Evidence for such effects has been found in studies [178] of isolated proteins with the β-MHC mutations R249Q and H251N, which are in the head–tail interaction region on the myosin head, as well as D382Y, P710R and R719W in the head–head interaction regions. Production of hypercontractility by destabilization of the IHM was also later demonstrated for the R403Q and R633H mutations in β-MHC [191]. The hypothesis for hypercontractility by destabilization of the IHM also gains support from a number of other studies. First, whereas only a rather small fraction of the β-MHC HCM mutations are predicted, based on optimized molecular models, to directly affect the IHM, a vast majority of the mutations may destabilize the IHM by indirect mechanisms [26].

To summarize, the studies of isolated proteins generally corroborate the idea of hypercontractility as a direct effect of HCM mutations in sarcomere protein genes, either through modified actomyosin kinetics, destabilization of the IHM or increased Ca-sensitivity. It is important to note, in this connection, that increased Ca-sensitivity would be expected not only to slow relaxation and cause diastolic dysfunction but also to increase the twitch force even without other changes due to low level of activation during the cardiac twitch under normal physiological conditions [16]. These findings are consistent with increased ATP consumption both during the cardiac twitch and, to different degrees, in relaxed muscle. They may thus be consistent with the evidence for energy depletion in the HCM. However, it is important to note that some counter-examples and inconsistent experimental findings raise doubts about the idea that all sarcomere mutations cause hypercontractility. In addition to some cases mentioned above, the atypical R243H mutation in switch I of the β-MHC is of interest. First, it is in a critical position of myosin for effective catalysis [192]. Second, it cannot readily be associated with destabilization of the IHM [26], and third, it has been associated with both apical HCM and dilated cardiomyopathy [26].

#### 4.4.2. Studies on Cells and Myofibrils Isolated from Adult Human Muscle

Whereas the results from isolated proteins seem to quite broadly support the hypercontractility hypothesis (with some exceptions), the evidence from myofibrillar preparations seem, at first glance, to argue against this idea. Thus, the maximum isometric force is generally reduced in myofibrils isolated from patients (reviewed in Reference [48]), effects that are associated with reduced phosphorylation levels of troponin I and cMyBP-C [48,86,193]. This seems to apply with most HCM-causing mutations whether in β-MHC, cMyBP-C or thin filament proteins [171,194,195,196]. However, there are a number of issues to consider in relation to these findings. First, the studied myofibrils are obtained from hearts subjected to HCM remodeling, which could partly explain apparent differences from results using isolated proteins, primarily expressed in cell systems. Second, the maximum isometric force at full calcium activation is not representative of the force produced during a cardiac twitch because the level of calcium activation during the twitch is far below saturation. Thus, an increase in calcium sensitivity or disturbed calcium handling with increased calcium concentrations in the presence of HCM mutations may substantially increase the twitch force despite reduced force at maximum calcium activation. This is clearly illustrated in a comparison of myofibril maximum force and twitch force in a mouse model where the latter force level was increased 3-to-4-fold by the mutation with no change in the maximum calcium-activated force of myofibrils [142].

The calcium sensitivity is usually increased in myofibrils (or skinned fibers) isolated from patients and in such preparations reconstituted with mutated thin filament proteins, expressed in cell systems [28,164]. The latter findings suggest that the increased Ca-sensitivity is a primary effect of the mutations. However, other studies instead show evidence for hypocontractility. Thus, fibers isolated from the soleus muscle, expressing the β-MHC G741R or R403Q mutations, exhibited decreased maximum velocity of shortening and decreased isometric force generation [93]. Similar results were found in another study of the R403Q mutation where isometric tension production was lower [197]. Yet another study showed that fibers isolated from human soleus muscles carrying the R719W or the R723G β-MHC mutation, exhibited an average reduction in calcium sensitivity, albeit with large variability among individual fibers [28]. Similar results were found in human cardiac cellular preparations and isolated myofibrils [194] (reviewed among other similar findings in Reference [48]). Thus, maximal force generating capacity per myofibril cross-sectional area was reduced with various β-MHC mutations (R403Q, R442C, V606M, R663C, R694C, S782N, R858P, R869H and T1377M) in experiments at submaximal calcium levels.

A confounding factor related to the mutation effects on calcium sensitivity in myofibrils and skinned fibers from patients is that they are often reversed if the reduced TnI phosphorylation levels [193] are restored to normal by treatment with protein kinase A [171]. This result would seem to suggest that the increased Ca-sensitivity is a secondary rather than a primary effect. However, there are additional levels of complexity, as suggested by in vitro motility assay studies using thin filaments reconstituted with troponin from HCM patients. As further discussed above, it was found that these filaments, despite appreciably lower TnI phosphorylation levels than in troponin from donor hearts, exhibited the same Ca-sensitivity in the in vitro motility assay, suggesting an uncoupling between the phosphorylation level and Ca-sensitivity in HCM [186]. This effect does not seem, however, to have been convincingly reproduced by using other experimental systems than in vitro motility assays (cf. Reference [48]). In myofibrils, for instance, increased TnI phosphorylation levels instead often restore the Ca-sensitivity toward normal as discussed above.

A reduced maximum isometric force at full calcium activation is consistent with increased rate of actomyosin cross-bridge detachment, because this would reduce the time spent by cross-bridges in force-generating states. The idea of increased cross-bridge detachment rate with several HCM mutations is also consistent with increased rate of relaxation found in myofibrils affected by different HCM mutations after isometric contractions at full calcium-activation (see References [81,130,195]; reviewed in References [21,48]). Interestingly, this effect would also increase the ATP turnover rate during isometric contraction and other contractions at high tension levels where the ATP turnover rate is limited by the cross-bridge detachment rate [21,114]. These tension levels are those where the heart operates during a large fraction of its working cycle. This includes isovolumetric systolic contraction and relaxation, as well as a substantial fraction of the ejection phase where velocity is, on average, ~1/3 of the maximum velocity [16]. Under these conditions, a fast cross-bridge detachment rate would cause inefficient contraction with high energetic cost.

It is important to note that slowed relaxation in intact cardiac muscle, consistent with diastolic dysfunction, is not contradictory to the findings of faster relaxation of myofibrils at full calcium activation. Despite the latter effect, relaxation can be appreciably slowed during the cardiac twitch (with submaximal calcium activation) either due to increased calcium sensitivity of the thin and thick filament regulatory machinery (e.g., associated with changes in troponin–tropomyosin, cMyBP-C and myosin RLC) or slower calcium pumping, e.g., inhibition of the SR calcium pump due to energy depletion.

While the dominant picture is that HCM-causing mutations increase the calcium-sensitivity, there are exceptions. Some of these, which are associated with the β-MHC mutations, were considered above. Another example of a mutation that has been found to decrease calcium-activated tension and stiffness and reduce myofilament Ca^2+^ sensitivity is the E22K mutation in the RLC of myosin [172]. This mutation causes HCM in both mice and humans. Whereas the mentioned study was performed in skinned papillary muscles from transgenic mice, the RLC was of human type, and similar HCM disease phenotypes are seen in mice and humans. However, it is important to note that there are variable results for the effect of the RLC E22K mutation on Ca^2+^ sensitivity [198,199], with one of the studies [198,199] even showing an increased Ca-sensitivity (discussed by Zhang et al. [172]). One further example of hypocontractility in cell and myofibril studies has been seen with the HCM-causing actin mutation A331P that decreased both the maximum isometric tension and calcium sensitivity in skinned bovine muscle fibers that were reconstituted with actin expressed in insect cells [164]. Importantly, however, opposite effects, i.e., reduced Ca-sensitivity, were observed with other HCM-causing actin mutations in a similar experimental system [163].

To summarize, the myofibril and skinned fiber results suggest that the maximum isometric force is usually reduced at full calcium activation with a range of HCM-causing mutations. This is generally associated with faster actomyosin cross-bridge detachment and increased ATP turnover rate (consistent with poor energy efficiency). Furthermore, the Ca-sensitivity of contraction is mostly increased, but it is not entirely clear whether this represents primary effects of the mutations, because the effects of reduced TnI phosphorylation complicate the interpretation. Independent of the reason for increased Ca-sensitivity, this effect would be compatible with slower relaxation and increased resting tension. However, importantly, there are exceptions from the overall picture, and HCM mutations in β-MHC, actin and the myosin RLC have been reported to be associated with reduced instead of increased Ca-sensitivity.

#### 4.4.3. Studies Using Cardiomyocytes and Cardiac Tissue Engineered from Induced Pluripotent Stem Cells

Studies using engineered cardiac cells and tissues of human origin give a variable picture with regard to hypercontractility and hypocontractility in HCM. This is suggested by a recent review of earlier works [49], and the view is supported by findings in more recent studies that we consider here. Whereas several of these studies report increased twitch forces/cell shortening and/or increased calcium sensitivity for HCM mutations in β-MHC [27,133,179], TnT [134,188,200] actin [32] and cMyBP-C [27,133], this is not consistently found. For example, in one study, Bhagwan [32] (Figure 6) reported increased twitch force, compared to the wild type, of engineered human cardiac tissue for the actin E99K mutant but *reduced* force for the β-MHC R453C mutant. The latter effect was studied with both heterozygous and homozygous mutations and more severe effects were found in the latter case. Both the cells with the actin E99K and the β-MHC R453C mutation [32] exhibited typical HCM characteristics, including increased cell volume and sarcomere disarray. The reduced twitch force in engineered cardiac tissue, seen with the β-MHC R453C mutation [32], reflects both actomyosin cross-bridge kinetics and the level of calcium activation. The result therefore accords with the view that neither cross-bridge kinetics nor Ca-sensitivity is affected in hypercontractile direction by the mutation. The relaxation after contraction is prolonged in all our reviewed studies of engineered cardiomyocytes [27,32,133,200], except in one study of a TnT R286H mutation [134].

To summarize, the studies of engineered human cells and tissues based on hiPSC give a variable picture with regard to the existence of hypercontractility in HCM, but they generally suggest prolonged relaxation and invariably increased energy usage [27,32,133,134]. Thus, in all of these studies, the energy usage was significantly increased by the presence of the HCM mutations, whether twitch force/cell shortening was increased or not. In some studies [32,133], these effects were explicitly correlated with secondary changes, such as the accumulation of reactive oxygen species (ROS) and initiation of certain cell-signaling pathways.

#### 4.4.4. Whole Hearts and Cardiac Imaging

Studies of whole hearts of HCM mutation carriers (individuals with a HCM genotype but without ventricular hypertrophy; G+ LVH-), using imaging, are of value for elucidating the early initiating events that later lead to the overt HCM phenotype. One method that has been quite extensively used in this connection is the echocardiography-based tissue Doppler imaging (TDI) technique. It was found in a pioneering study by Ho et al. [25] that early diastolic myocardial velocities (Ea velocities) were markedly reduced in mutation carriers, thus showing evidence for diastolic dysfunction. Later studies reported somewhat variable results, e.g., some studies reported no clear differences in TDI-based assessment of diastolic function between mutation carriers and a control group [201,202]. More recently, a meta-analysis [173] involving the three mentioned studies and seven more suggested statistically significantly lower Ea value at the septum in mutation carriers than among individuals in the control group implying that mutation carriers generally exhibit diastolic dysfunction. The latter effect would be compatible with often (but not invariably) increased calcium-sensitivity of contraction (see above). However, in the whole heart, diastolic dysfunction may also be worsened by interstitial fibrosis which could be present not only in fulminant HCM but also in mutation carriers [203].

Most imaging studies report unaltered or increased ejection fraction in mutation carriers, suggesting that systolic function is not affected. However, it has been argued [30,139] that the ejection fraction is a blunt instrument in assessing systolic function. If measures such as absolute values of end-systolic and end-diastolic volumes [30], as well as global myocardial longitudinal and circumferential strain parameters [30,137], are instead considered, the evidence seems to favor changes in systolic function also in mutation carriers. Other studies, however, have shown only minor changes in global strain parameters and, instead, using speckle-tracking echocardiography, found regional differences in myocardial strain [166,202] with mechanical dyssynchrony associated with altered degree of rotation and twist of the heart during systole [31,166]. The regional differences in strain parameters indicate subtle disturbances in systolic function with force imbalances between ventricular wall segments. This has, for example, been attributed to heterogeneities in contractile strength between subendocardial and sub-epicardial fibers, possibly due to more extensive myofibrillar disarray subendocardially [166]. However, also other possibilities exist, including inhomogeneous myocardial activation caused by conduction disturbances [137] or exacerbation, in the presence of HCM mutations, of heterogeneities [204,205,206] existing already in the healthy heart (reviewed in Reference [35]). In addition, contractile inhomogeneities could emerge due to structural abnormalities, e.g., in the form of myocardial crypts observed in mutation carriers [207].

In relation to energy depletion and poor energy efficiency [11], Crilley et al. [167], using phosphorus-31 magnetic resonance spectroscopy, found that the [Phosphocreatine (PCr)]/[ATP] ratio at rest was increased both in phenotype positive and phenotype negative individuals (mutation carriers) with HCM mutations in three different genes. Further evidence in support of inefficient energy usage (low work output per O_2_ consumption) in HCM mutation carriers have been provided by using [11C]-acetate positron emission tomography (PET) and magnetic resonance imaging [80,168]. Evidence for severe metabolic disturbances was also provided by a multi-omics study [154] applied to human septal myectomy samples from HCM patients (with ejection fraction >55%) compared to donor hearts. These findings, suggesting “perturbed metabolic signaling and mitochondrial dysfunction”, are consistent with ideas [11,13,208], emphasizing the importance of energy deficiency and dysfunctional mitochondria in HCM pathogenesis.

The systolic and diastolic functional parameters from cardiac imaging, seen in mutation carriers, seem to be generally similar but deteriorate further in patients with a fully developed HCM phenotype [30]. Accordingly, Serri et al. [209] and Shetty et al. [210] found reduction of both longitudinal, circumferential and radial strain in HCM despite normal ejection fraction. Shetty et al. [210] also found increased torsion of the heart. Another study [211] has shown abnormal myocardial deformations during systole in children with HCM, effects that are more severe for more extensive hypertrophy. With regards to energy depletion, Guclu et al. [80] (see above) found that the ratio between external work and myocardial oxygen consumption was highest in healthy individuals, intermediate in mutation carriers and lowest in individuals with fully developed HCM. Complementing this information, Dass et al. [212] observed that the increased [PCr]/[ATP] ratio in HCM patients was further exacerbated during exercise. In addition, studies of oxygen consumption and mechanical efficiency using PET scans and magnetic resonance imaging suggest reduced oxygen consumption with further reduced mechanical efficiency upon progression from phenotype-negative to phenotype-positive HCM. It is important to note that, in patients with fully developed disease, both energy depletion and contractile disturbances, e.g., non-uniform force-generating capacity between segments, may be partly due to secondary changes unrelated to properties of the individual cardiomyocytes. This includes the effects of compromised blood supply due to small vessel disease in HCM (reviewed in Reference [13]) and effects of the thicker myocardial wall per se. Although this is not likely to be an important early event (cf. Reference [168]), it would make the situation worse in fully developed disease, particularly during physical exertion.

### 4.5. Hypercontractility vs. Hypocontractility—Overall Summary

Hypercontractility on the molecular level, in one form or another (increased power output by a single motor, destabilization of the IHM and/or increased Ca-sensitivity), seems to result from a large fraction of the HCM-causing mutations independent of the gene. However, the lack of systematic studies of each individual mutation on different hierarchical organization levels overall weakens the arguments for the hypercontractility hypothesis, as evidence instead supporting hypocontractility can be found on at least some hierarchical levels for several mutations. Strikingly, it is difficult to find a single mutation where consistent evidence has been reported for all (examined) experimental levels. Again, this could mean lack of systematic studies, but there clearly seem to exist contradicting findings pointing toward either of the hypotheses in Table 1 for most specific mutations. The weakness of findings limited to experimental results from one given hierarchical level is illustrated by comparing recent findings from studies of S1-like and HMM-like myosin constructs. As summarized above, hypocontractility on the S1 level, with a given HCM mutation, is in several cases transformed into hypercontractility for the HMM-like constructs due to reduced population of SRX states. Clearly, one cannot overlook the possibility that similar, but possibly opposite, changes occur when going from HMM-constructs to half-sarcomeres, and further to entire myofibrils (with large numbers of sarcomeres in series), muscle fibers and whole hearts [17]. This possibility is demonstrated for the R453C β-MHC mutation by comparison of studies of S1, showing hypercontractility with respect to force-producing capacity [99,174], with studies of engineered cardiomyocytes and tissue [99], showing hypocontractility (Figure 6). However, here one must be aware of complexities, e.g., possible immaturity and/or secondary tissue remodeling, including posttranslational modifications, in the engineered cells (see above).

Generally, the evidence for hypercontractility seems to be less consistent on the hierarchical levels above isolated proteins: from myofibrils over cardiomyocytes (isolated or engineered) to the whole heart. The basis for these findings needs to be understood. A speculative idea could be that the primary effect of a mutation (possibly with some exceptions) is indeed hypercontractility and that compensatory mechanisms of the muscle or secondary effects give rise to hypocontractility. One reason could be hypocontractility due to non-uniformities, as discussed below. Alternatively, hypocontractility might arise as a consequence of mechanisms aimed at compensating the hypercontractility, e.g., dispersing the extra force generated by increasing cardiomyocyte size by diluted myofibril density (cf. Reference [194]) and/or post-translational modifications that decrease force generation.

Strikingly, as opposed to somewhat conflicting views regarding the predominance of hypocontractility vs. hypercontractility, all studies evaluated here, independent of hierarchical level, show results consistent with inefficient energy usage in HCM (cf. Reference [11]). This effect therefore seems to be a strong candidate for an early event downstream to the initiating events (e.g., hypo/hypercontractility and contractile instabilities) in stimulation of pathologic remodeling.

### 4.6. Hypothesis 3: Non-Uniformities as Primary Change

#### 4.6.1. Origin of the Hypothesis

Cardiac-imaging studies demonstrate functional non-uniformities over the ventricular wall in HCM mutation carriers [31,166,213], making the “non-uniformity” or “contractile imbalance hypothesis” [28,33,34,35,36,39,153] highly relevant. This hypothesis can, however, be traced back to an experimental system on a different hierarchical level, demonstrating that single cells from given patients [28] generate highly different force levels at identical Ca^2+^ concentrations. Both β-MHC mutations giving an average decrease (R723G, A200V and R719W mutation) and increase (I736T) in calcium sensitivity were associated with appreciably increased variability in the force–pCa relationships between different muscle fiber segments isolated from a slow skeletal muscle from a given individual. Specifically, up to 10–20-fold differences between cells were seen for the β-MHC A200V and R723G mutations compared to 1.5-fold variation between cells in controls [28]. Such functional variability within the tissue may cause contractile imbalance where stronger cardiomyocytes over-contract, while weaker cardiomyocytes become over-stretched. These effects may disrupt the branched myocardial network to cause myocyte disarray and also activate profibrotic and hypertrophic signaling pathways [33,34,36,39,153] ideas that has also been put forward based on cardiac imaging results [214]. There have been several studies [28,33,34,37,39,40,59,153,162,215,216] suggesting that variability of the mutant-to-wild-type protein ratio among cells of the same tissue is the underlying cause of the imbalance. However, evidence has also been presented for a more severe phenotype if more mutated vs. wild-type protein (from β-MHC or cMyBP-C) is expressed on average [40,162,217,218].

Montag, Kraft and co-workers [37,39,119,153] focused on heterozygotes and associated “burst-like transcription” with “allelic imbalance”, while also adding a time component. The resulting mosaic of cells with varying strengths would, over time, be expected to further change the strength of the individual cells exacerbating the contractile imbalances. Due to the branching networks of the cardiac muscle, disruption of the cellular network would affect the heart more severely than similar non-uniformities in skeletal muscle [34]. Interestingly, the variability in expression levels of mutated and wild-type protein has not only been observed for β-MHC but also for cMyBP-C [219]. Whereas the evidence is rather strong for both contractile non-uniformities in HCM and variable expression levels of wild-type and mutated proteins, the causative relationship between the two phenomena is not equally clear, as we consider further below.

#### 4.6.2. The Non-Uniformity Hypothesis and Homozygote vs. Heterozygote HCM Mutations

Whereas differences in expression levels of mutated and wild-type sarcomere proteins between different parts of the myocardium may play roles in the pathogenesis, as suggested above, there are findings which question this idea. Particularly, if varying expression levels of mutant and wild-type protein would be of critical importance in pathogenesis, the disease would be expected to be appreciably more severe in heterozygotes (with expression of both mutated and wild-type protein) than homozygotes (expressing only proteins with HCM mutation). However, a recent study of pre- and post-natal mouse development showed myofibrillar disarray in both hetero- and homozygotes [135]. Another study showed that, although the hearts of mice with HCM mutations appeared normal at birth, the increase in size before reaching the hyperthrophy state was faster in homo- than in heterozygous mice [220]. Along the same lines, studies using hiPSC [32,47] reported more severe changes as a result of homozygous vs. heterozygous HCM causing mutations (Figure 6). Furthermore, there is evidence that the disease course in homozygous human patients is more severe than in heterozygotes [217,221,222,223,224,225]. In addition, the likelihood of a homozygous fetus or neonate to sustain life seems lower, which could contribute to explaining why only few homozygotes are found at adult stages [224,226] or that the homozygotes that make it to adult life have a mutation that causes milder disease.

To summarize, the available evidence suggests more severe remodeling of the heart, as well as more severe disease with a homozygous than a heterozygous mutation. Therefore, it seems that the “contractile imbalance hypothesis”, relying on different expression levels of mutated and wild-type protein in different cells, is limited by only accounting for HCM development in heterozygous individuals.

Nevertheless, the original idea pioneered by Kraft, Brenner and co-workers [28,33,34] that contractile imbalance is important in the HCM pathogenesis does not directly rely on the mechanism with different expression levels between cells of mutant and wild-type proteins [28]. Several other potential sources of heterogeneity could have similar effects, as discussed below.

#### 4.6.3. Possible Reasons for Non-Uniformities Other Than Varied Expression Levels of Mutant and Wild Type Alleles

The heterogeneities in the normal ventricle come in various flavors [35]. First, there are differences in the expression levels of different myosin isoforms with different mechanokinetic properties both over time and between different regions of the myocardium. During the early stages of embryonal development there is increased expression of embryonic (*MYH3 gene*) and neonatal-myosin (*MHY8*) that over time (weeks to months after birth) switches to primarily β-myosin (*MYH7*) in the human heart [227]. The two cardiac myosin isoforms in an adult human heart are the fast α-myosin (*MYH6*), which is predominant in the atria, and the slow and more energy-efficient β-myosin, which is expressed primarily in the ventricles [228,229]. The ratio between these two isoforms in the ventricle (α/β: ~1:10) has been implied to be crucial for optimal function [230,231,232,233]. Of particular interest in this regard is the difference between the endocardial and epicardial layers of the ventricle [206] with normally higher α-myosin expression levels in epicardial fibers (Figure 7A). Another difference between the epicardial and endocardial side of the myocardium is different fiber orientations [35]. In addition to the epicardial–endocardial gradients, there are differences in contractile strength between apical and basal parts of the heart [35,204], at least partly attributed to differences in phosphorylation levels (Figure 7B).

These differences, together with different timings of electrical activation and different delays [234,235,236], between electrical activation and force development may explain the normal variation in ventricular strength at a given time during the cardiac cycle between endocardial and epicardial muscle layers. For instance, in the normal heart, endocardial fibers contract before the epicardial fibers, which are then stretched. Such stretching may be more severe in HCM [35] (cf. Figure 7C) where stretch-activation (related to the Frank–Starling mechanism) is blunted (see below). In this connection, it is of appreciable interest to note that, in addition to a generally patchy hypertrophy in HCM, a gradient of functional deterioration has been observed between the subendothelial and subepicardial myocardium with the former being spared to a higher degree than the latter [213].

Most likely, there are also differences along the length of cardiac fibers/cells (possibly between neighboring sarcomeres) [230] in analogy with what has been observed along a given skeletal muscle fiber [237,238,239]. Indeed, similar local non-uniformities have been observed in cardiac myofibrils [240] where important physiological roles of the non-uniformities include the speeding up relaxation [237,241] and, possibly, smoothening of half-sarcomere contractions, when summed over the length of the muscle [242].

It is of interest to note that the existence of the normal variabilities, as outlined above, fits well into a “variation is function” theory [243], which implies that variation in the cell population inside a tissue is necessary for optimal function and that the loss of the variability may result in disease [244]. However, one may also turn this argument around, i.e., that the presence of variation may be deleterious in the case of altered function (even minor) of each individual cell, as in HCM. Thus, the normally existing fine-tuned heterogeneities along given muscle fibers [237,238,239] and over the myocardial wall [35,204] may become maladaptive and cause critical imbalances in the presence of HCM mutations. Several possible reasons may be considered as outlined below.

First, the possibility exists that the presence of the mutation somehow, by unknown mechanisms, triggers non-uniformities between different cells in the total expression level of the active mutated protein and that this variability in absolute expression level increases with the amount of mutated proteins. In such a case the variability between cells would be expected to be higher in homozygous than in heterozygous mutants.

Second, one can easily see that the fine-tuned differences in phosphorylation levels, fiber orientations and α/β-myosin ratio between different parts of the healthy myocardium may be compromised by disturbances in phosphorylation patterns, development of myofibrillar and cellular disarray and a reduced α/β-myosin ratio. The α/β-myosin ratio is also disturbed functionally when HCM mutations occur in β-myosin, which is generally the case.

Third, one may consider that the presence of HCM mutations exacerbate non-uniformities already in the embryonal stage. Thus, recent studies indicate that different degrees of myofibrillar disarray start in the early stages of life and affect the development of the heart tissue [63,135,245].

Fourth, the possibility has been considered [36,38] that the mutations lead to contractile instabilities by modifying actin–myosin interaction kinetics, such that each half-sarcomere exhibits an anomalous force–velocity relationship, with more than one metastable velocity for a given load. This could be deleterious in itself with sudden jumps in sarcomere length. It could also potentially enhance the effects of already existing non-uniformities [36] with local stretches and non-uniform contractions to the extent that these cause muscle damage or initiate remodeling by stimulation of specific cell-signaling pathways.

Finally, it is of interest to note that hypercontractility, suggested to be an outcome of several mutations (see above), may increase effects of already existing non-uniformities. Thus, even without other effects of the mutations, any absolute differences in force level between cells would be increased if the mutation increases force-production throughout (Table 3).

To summarize, one may consider the following factors as possible reasons for the emergence of deleterious contractile non-uniformities in HCM: (i) different expression levels between cells of mutant and wild-type protein (in heterozygotes), (ii) increased variability in total expression level between cells of the protein affected by the mutation, correlated with the level of mutated protein (most prominent in homozygotes), (iii) mutation-induced disturbances in the fine-tuned normal heterogeneities in the myocardium and (iv) exacerbated consequences of existing non-uniformities as a result of mutation-induced hypercontractility and/or mechanical instabilities with anomalous force–velocity relationship.

#### 4.6.4. Non-Uniformities, Cellular Adaptation and Energetics

In addition to some of the detrimental effects considered above, any contractile imbalance, independent of reason, may contribute to inefficient energy use by the cardiomyocytes; that is, if there is variability in strength along the cardiac wall, developed forces will stretch internal weak segments rather than producing effective shortening or tension in the entire ventricular wall. One may think that if some segment of the ventricular wall develops hypertrophy due to repeated excessive stretch, it may counteract the non-uniformity by gaining increased strength. However, alternatively, the possibility exists that the hypertrophied segment is further weakened due to compromised energy supply both due to diffusion problems and small vessel disturbances in HCM, as well lower myofibril density.

### 4.7. Perturbed Length Dependent Activation

Perturbed length dependent activation is not a central hypothesis that we consider. However, the phenomenon could influence development of HCM, possibly by modulating effects related to the three hypotheses considered above. The Frank–Starling mechanism of the heart is a feedback mechanism that causes increased stroke volume when the end-diastolic volume of the cardiac ventricle is increased. This is primarily believed to be due to increased calcium sensitivity of the contractile machinery upon increased length of the myocardial cells. As reviewed in References [24,48], this mechanism seems to be quite generally blunted in HCM (see also Reference [171]), primarily attributed to increased calcium sensitivity at shorter length with less room for further increase when the cell is stretched [24]. The effect seems to be only partially eliminated by Protein Kinase A mediated phosphorylation [48]. As pointed out in Reference [24], the blunted length dependent activation possibly contributes to diastolic dysfunction. However, we also note that the effect would be expected to reduce the mechanical stability of potential importance related to the non-uniformity hypothesis. Thus, if weak myocardial segments are stretched, a length-dependent increase in myofilament calcium-sensitivity (which seems to occur immediately during stretch [246]) would counteract this effect. Clearly, if this length-dependent increase is blunted in HCM (for whatever reason), it would weaken one important mechanism that confers mechanical stability. We are aware that there is much more to be said about this mechanism. However, here, we only note its potential importance and refer the reader to the abovementioned reviews for more details. The potential importance of a blunted length-dependent activation in exacerbating effects of contractile non-uniformities is nicely illustrated by a multiscale electromechanical model of the heart [247].

## 5. Cell Signaling and Pathologic Remodeling

Several cell types are involved in the signaling pathways in HCM that lead to cardiac remodeling [74,248]. However, cardiomyocytes and fibroblasts dominate, and we here focus on the cardiomyocytes. With regard to hypertrophy, the signaling paths are distinct in physiological (e.g., in response to exercise) and pathologic processes [74,249]. In the former conditions, hypertrophy has long-term adaptive roles. In pathologies, on the other hand, hypertrophy may be adaptive in the short term but maladaptive in a longer perspective. Causes of pathologic hypertrophic remodeling, in addition to HCM, include volume overload, as in mitral regurgitation, or pressure overload as in hypertension and aortic stenosis but also partial myocardial destruction as in myocardial infarction. In contrast to physiological hypertrophy, pathologic hypertrophy is, to different degrees, associated with mitochondrial dysfunction, insufficient angiogenesis, cell death and fibrosis, as well as changed patterns of posttranslational modifications and protein expression (to a more fetal type). Interestingly, the changes in the proteoform seem to be similar independent of the genetic basis of the HCM [250]. Here, we consider the key signaling pathways. More general insight and with appreciably more details can be found in the excellent reviews [74,248,249] that we primarily cite here.

Increased intracellular calcium concentration, increased cellular tension per se and metabolic disturbances (e.g., accumulation of ROS and increased PCr/ATP ratio) seem to play central roles in the signaling pathways that lead to pathologic hypertrophy and associated phenomena (e.g., fibrosis and gene reprogramming) whether in HCM or other pathologic conditions with hypertrophy [74]. This is consistent with the results presented above, particularly from studies using hiPSC-derived cardiomyocytes which specifically pinpoint metabolic factors (e.g., References [32,133]; see also Reference [11]) and calcium-related signaling [32] as early steps in the HCM remodeling. Most likely, these effects are not far downstream in the signaling pathway from the primary mutation-induced functional change. With regard to increased calcium levels, these may be a result of mechanosensing mechanisms or Angiotensin II (ATII) signaling where ATII may be acting as a hormone or a paracrine or autocrine agent [249,251,252]. As such, it has been found to be released by the cardiomyocytes themselves as consequence of locally increased tension or stretch [253]. Increased intracellular Ca-levels under some phases of the cardiac cycle could also result because of increased Ca-buffering by troponin, due to HCM mutations with increased Ca-sensitivity. However, it could also be secondary to energy depletion that inhibits cardiac calcium pumps (Figure 2). Increased calcium concentrations act downstream, e.g., by binding to calcineurin that changes gene transcription by dephosphorylating nuclear factor of activated T cells (NFAT) to stimulate its transfer to the nucleus [74] (see also Reference [32]). Calcium also binds to calmodulin, activating, for example, the calmodulin-dependent kinase, which, among other effects, causes loss of hypertrophy-inhibiting nuclear class II histone deacetylase 4 (HDAC4) from the nucleus [74]. However, not all HCM mutations seem to be associated with altered Ca-signaling [32,133,142].

In addition to calcium-centered mechanisms, the accumulation of ROS, ADP and/or other metabolites [80,133,167,254] could be key factors in initiating the pathologic signaling. Additionally, the accumulation of metabolites causes mitochondrial dysfunction with secondary cell death.

Other alternative mechanisms could be direct effects of mechanosensing. A priori, this could result either from hypercontractility with higher tension under certain conditions, hypocontractility with lower tension or, finally, by local stretch with very high tension in some cells according to the non-uniformity hypothesis. Increased tension levels, particularly in stretch, could, as pointed out above, lead to the local release of ATII paracrine and autocrine effects on cardiomyocytes, e.g., causing increased intracellular calcium concentration but also calcium independent activation of MAPK–MEK pathways with altered gene transcription as basis for remodeling [74]. Additionally, ATII has also been found to have profibrotic effects on cardiac fibroblasts [252]. Finally, in relation to ATII, AT1-receptors (for ATII) seem to be directly sensitive to mechanical stress [255]. In this connection, it is of interest to note that there are several mechanosensing systems in the cardiomyocyte that may lead to release of ATII in addition to several other downstream signaling events. Mechanosensor molecules include [248,256] mechanically gated ion channels but also other proteins on the cell-surface associated with the costameres (e.g., integrins, vinculin and dystrophin) and in the intercalated discs between individual myocytes (e.g., catenins and vinculin). However, there are also mechanosensing proteins in the sarcomere, e.g., in the Z-lines [257] and in parts of the giant protein titin between the thin and thick filaments. For instance, a stretch of critical titin domains, possibly involving binding of the four-and-a-half LIM domains protein 1 (FHL1), initiates MAPK hypertrophic signaling pathways with downstream activation/effects on MEK2 and ERK2.

Finally, substantial increases in local tension levels could cause cell destruction by mechanical forces, particularly during stretch, which in itself could result in cell death and scarring.

## 6. Clues from HCM Causing Mutations in Proteins outside Myofilaments

The idea that inefficient energy usage is central in initiating the pathologic remodeling in HCM is consistent with findings that hereditary disturbances of energy-producing mechanisms lead to histological, anatomical and clinical phenotypes that are virtually indistinguishable from HCM due to sarcomeric protein mutations [11,13,45]. One example includes Friedreich’s Ataxia (FRDA) caused by loss of the protein frataxin, which is a mitochondrial protein involved in energy homeostasis [258].

The ideas that mechanical disturbances associated with hypercontractility (independent of energy status), hypocontractility or non-uniformities could stimulate signaling pathways to produce HCM phenotypes is consistent with effects of some mutations in mechanosensing proteins. For instance, mutations in the glycine-rich protein 3 (also known as cardiac muscle LIM protein), as well as the binding partner telethonin in the Z-line, have been associated with human HCM [259]. This protein complex has important roles in mechanosensing and mechanotransduction.

There is also evidence from hereditary changes of proteins outside the myofilaments for involvement of Ca^2+^-based signaling in the early stages of HCM. Thus, Noonan syndrome, with multiple lentigines due to mutations in the non-receptor protein tyrosine phosphatase SHP2, includes an HCM-like phenotype that has been attributed to augmented intracellular Ca^2+^ cycling and increased number of power-generating sarcomeres [260].

## 7. A Unified Hypothesis

In our review of the literature, we discerned a picture where HCM mutations often cause various forms of hypercontractility as a primary event. Importantly, however, there are exceptions to this picture that need to be further investigated. In contrast, we could not find any exceptions to the picture with a disturbed energy efficiency in HCM, an effect that seems to be important already before development of the characteristic disease phenotype. This is consistent with early suggestions [11] and with findings that several mutations in ATP producing systems [13] also lead to HCM with quite similar phenotype as caused by sarcomere protein mutations. However, the evidence is not 100% convincing that hypercontractility in its different forms is always the primary factor behind energy depletion. Whereas hypercontractility may often be part of the problem, this may not always be the case, or it may require the simultaneous existence of other similarly important factors. Therefore, we propose, that hypercontractility or hypocontractility, in association with different forms of contractile non-uniformities promote further non-uniformities (Figure 8) leading both to energy depletion and mechanical disturbances. The latter, in turn, cause cell damage and activate various signaling pathways that lead to remodeling with hypertrophy, further myofibrillar and cellular disarray and fibrosis.

## 8. Identification of the Primary Mutation-Induced Functional Change That Initiates Remodeling—A Case for Studies Using Isolated Proteins

One major difficulty in studies aiming to identify primary mutation-induced changes behind pathologic remodeling is that most experimental preparations of human origin have been subjected to one or other form of remodeling. Necessarily, this affects functional studies, e.g., muscle mechanics, and would blur primary effects. Whereas remodeling effects such as reduced myofibrillar density (cf. Reference [194]) and myofibrillar disarray are eliminated by the use of myofibrils for mechanical studies, the effects of posttranslational modifications are not. The latter problem also applies if isolated proteins are obtained from human heart tissue. Furthermore, in the cases of all the mentioned preparations, there is generally a poorly characterized mixture of wild-type and mutated protein with varying phosphorylation levels. There have been claims that engineered human cells and tissues circumvent these problems on the assumptions that they have properties that reflect the primary mutation-induced effect. However, as pointed out above, the signs of hypertrophy and myofibrillar disarray in these preparations raise concern in this regard. Thus, these effects, particularly hypertrophy, are not evident in the hearts of mutation carriers. Therefore, it seems unclear whether the engineered tissue exhibits primary mutation-induced changes or has, instead, undergone fast remodeling events to exhibit characteristics of fulminant HCM.

Notwithstanding, if the purpose of an investigation is to identify primary mutation-induced changes, it seems necessary to look at a level where this emerges most clearly. The obvious answer here seems to be the use of single molecules expressed in cellular systems where the proteins have not been subjected to pathologic remodeling effects. If one can ensure that the proteins are properly folded and that essential (but not pathological) posttranslational modifications are present, the single molecules would have appreciable advantages. However, an important lesson in this context is taught by the different effects of given β-MHC mutations on the function of isolated single motor domains (myosin S1), two-headed HMM, full-length myosin and/or myosin filaments/thick filaments. These findings suggest that it is important to consistently perform studies on different levels of hierarchical organization. With regard to the effects of mutations on the IHM, these are revealed by studies that extend from the single motor domain to two-headed single myosin molecules (see above) and myosin filament preparations [261]. However, the possibility exists that further complexities of the mutation effects would be revealed by studies of motor ensembles on a range of different levels of hierarchical organization (cf. Reference [124]).

Higher levels of hierarchical organization could, most simply, constitute interacting thin and thick filaments. However, it would also be of interest to reconstruct 3D ordered mini-sarcomeres from isolated proteins. Whereas multiscale computational models [104,179] have similar goals, models are always approximations that might miss key aspects of the complex biological system. Here, it would instead be ideal to use isolated proteins of human cardiac origin, including the protein with the mutation of interest, all expressed in cell systems for better control of secondary remodeling effects. With regard to assembly of ordered arrays of proteins from the bottom up, using isolated proteins, and recording of functional properties, this is already possible when it comes to single thick and thin filaments [262,263,264]. It would be of interest to expand the experimental systems to 3D ordered arrays, which may then be directly compared to myofibrils isolated from cells to first identify and then clarify the role of any HCM related posttranslational modifications.

## 9. Conclusions and Perspectives

Current knowledge of the HCM pathogenesis relies on experimental studies using a range of disease models, from single molecules and isolated protein ensembles to myofibrils and cellular preparations (including engineered cells). Important information is also derived from cardiac imaging, as well as studies of human heart samples from surgery, etc. Our overview suggests the importance of comparing results from different hierarchical levels for full insight. The focus of this review was on functional assays; however, we refer to other recent papers [74,248,250] when it comes to important omics studies to evaluate changes in protein expression patterns, patterns of posttranslational modifications, etc.

Studies on different hierarchical levels support the idea that hypercontractility is a central, primary effect of most HCM causing mutations. However, there are exceptions that need to be considered. Moreover, contractile imbalance may be a primary effect that is associated with a range of HCM mutations. Together with hypercontractility, this effect may contribute to the inefficient energy usage, which seems to be unequivocally observed in HCM. Here, downstream disturbances in mitochondrial function and metabolic pathways [13,154,208] may exacerbate the situation. Furthermore, both hypercontractility and contractile imbalance (with locally high tension) may stimulate mechanotransduction-initiated signaling pathways in cardiomyocytes (cf. Reference [36]), leading to hypertrophy and fibrosis. Finally, both high tension levels and energy depletion may cause cell death per se, with subsequent scarring.

Some major remaining questions that we identify include the following: (i) Is energy depletion and inefficient energy usage a consequence of all HCM causing mutation, i.e., can counter-examples be found to refute this idea? (ii) Are the effects under Question (i) primarily attributed to hypercontractility or are also hypocontractility or non-uniformity major factors? (iii) Can myofibrillar disarray be viewed as a primary mutation-induced effect in the infant heart, i.e., are mutations producing the effect already during embryogenesis? (iv) In the latter case, why do the mutations cause such effects? Is it due to certain changes in the force-producing mechanism, e.g., different forms of hypercontractility or non-uniformity during muscle and sarcomere development? (v) To what extent are changes in hiPSCs upon HCM mutations representative of the primary mutation-induced change(s)? (vi) Is the blunted length dependent activation in HCM secondary to hypercontractility, e.g., does an increased calcium sensitivity in HCM or an increased ratio between myosin heads in SRX/DRX states prevent further increases upon increased activation?

The hypocontractility versus hypercontractility dilemma is, by itself, telling us that HCM has highly complex pathogenesis. One issue that may contribute to the difficulties is the use of experimental protocols that differ, even in details, between labs. In order to efficiently tackle pathophysiologic mechanisms of such a complex disease, open and strong collaborations between scientists may be essential. This would, of course, mean the introduction of a complete “open science” approach, and the use of identical standard operating procedures for each experimental task, (e.g., IVMAs, protein purifications, etc.), similar to, for example, the OECD Good Cell Culture Practice document [265].

## Figures and Tables

**Figure 1 ijms-23-02195-f001:**
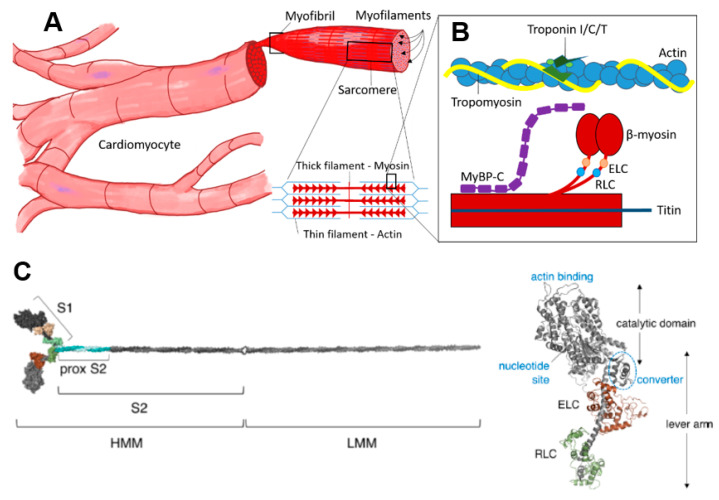
Cardiac contractile machinery from cells to proteins. (**A**) Cardiomyocytes, myofibrils and schematic of sarcomeric structure with thick and thin filaments. (**B**) Schematic showing key sarcomere proteins and their relative organization. (**C**) Molecular model of isolated myosin II molecule modified from Reference [15] under a CC-BY license (http://creativecommons.org/licenses/BY/4.0/) accessed on 18 January 2022. Different well-defined myosin fragments that are used in various studies are indicated: S1, subfragment 1; S2, subfragment 2; HMM, heavy meromyosin and LMM, light meromyosin. The myosin motor domain, S1, is shown in greater detail to the right, with the essential (ELC) and regulatory light chain (RLC) indicated in brown and green, respectively.

**Figure 2 ijms-23-02195-f002:**
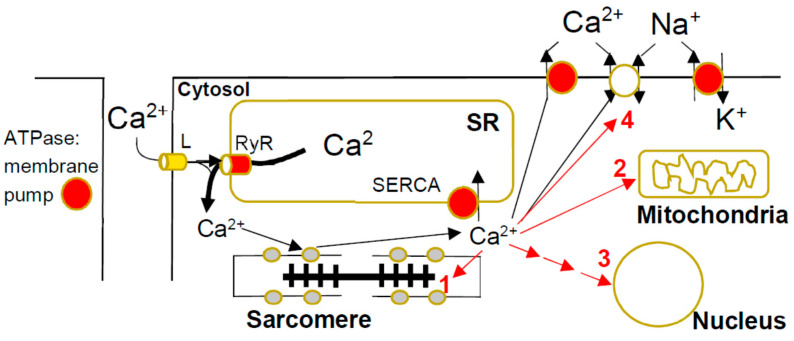
Calcium metabolism in cardiac myocyte and HCM-induced disturbances. The calcium ions enter during the cardiac action potential through L-type (L) Ca^2+^-channels in the T-tubule membrane and release larger amounts of Ca^2+^ from the sarcoplasmic reticulum (SR). This occurs through the binding of Ca^2+^ to the ryanodine receptor with opening of the integral Ca^2+^ channel. The resulting increase in [Ca^2+^] in the cytosol ([Ca^2+^]_c_) leads to Ca^2+^-binding to troponin on the thin filaments with contractile activation. The [Ca^2+^]_c_ is then lowered back to resting values by active pumping into the SR by the SR calcium pump (SERCA), as well as active pumping to the extracellular fluid and exchange with Na^+^ driven by the Na^+^-gradient built up by the Na/K ATPase. In the presence of HCM mutations, [Ca^2+^]_c_ may be increased by intracellular Ca^2+^ buffering due to increased binding to troponin and reduced pumping into the SR, as well as out of the cell, due to inhibition of active pumping by energy depletion. The increased [Ca^2+^]_c_ can exert negative effects in HCM by (1) stimulation of phosphorylation of sarcomere proteins, (2) direct effects on mitochondrial function [79], (3) activation of Ca^2+^-dependent signaling (see text below) and (4) arrythmogenicity.

**Figure 3 ijms-23-02195-f003:**
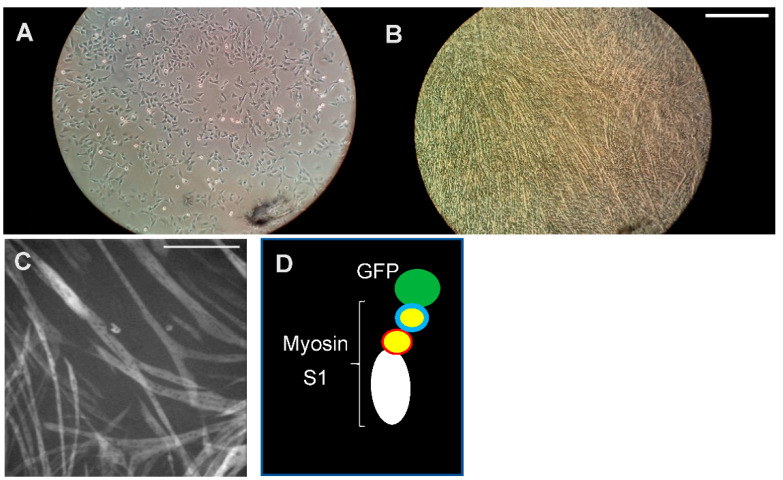
Myosin expression. Cardiac myosin heavy chain (β-MHC) needs to be expressed in mammalian muscle cells for correct folding and full activity. (**A**) Mouse myoblasts (C2C12) seeded in a well of a multi-well plastic plate. (**B**) The cell culture well after 7 days of differentiation of the cells into elongated myotubes in which fully functional myosin motor domains can be produced. Scale bar: 1 mm. (**C**) Expression of myosin S1 motor fragments by myotubes as reported by the GFP-tag fluorescence. Scale bar: 0.5 mm. (**D**) Schematic illustration of expressed S1-GFP construct.

**Figure 4 ijms-23-02195-f004:**
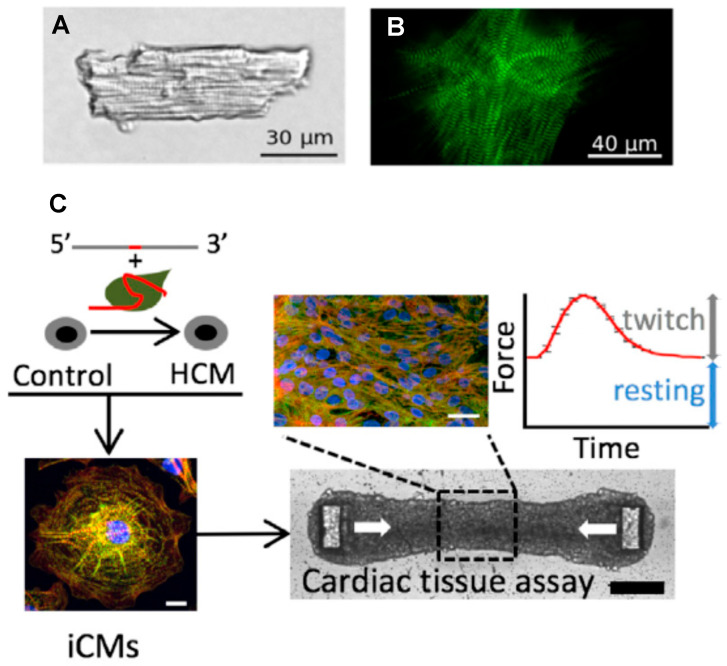
Adult cardiomyocyte vs. myocytes and cardiac tissues derived from hiPSC. (**A**) Cardiomyocyte isolated from adult mouse heart. (**B**) HiPSC-derived cardiomyocyte. (**C**) Generation of an isogenic HCM model by first (top left) using the CRISPR/Cas9 system to introduce specific β-myosin and cMyBP-C mutations on an isogenic background and then stimulating differentiation to cardiomyocytes (iCMs), followed by mixing with fibroblasts and extracellular matrix to form cardiac tissue, forced into a geometry that allows for a convenient recording of twitch force development upon electrical stimulation (top right). Panels (**A**,**B**) reproduced from Reference [27], and panel (**C**) from Reference [133], all under a CC-BY license (http://creativecommons.org/licenses/BY/4.0/) accessed on 18 January 2022.

**Figure 5 ijms-23-02195-f005:**
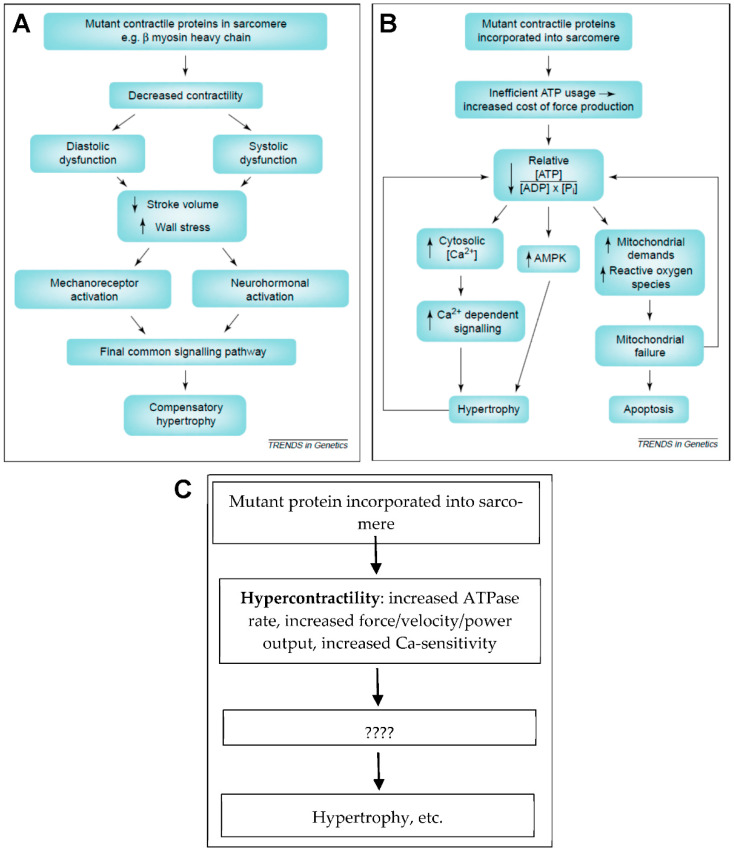
Hypocontractility, hypercontractility and energy depletion/energy inefficiency. (**A**) Hypocontractility hypothesis as presented (and partly refuted) in Reference [11]. (**B**) The hypothesis presented in Reference [11] that inefficient energy usage with energy depletion leads to pathologic remodeling. (**C**) Hypercontractility hypothesis where hypercontractility, in different forms as indicated, represents the primary mutation-induced functional change that, via secondary effects (????; e.g., energy depletion), leads to pathologic remodeling of the left ventricle. Panels (**A**,**B**) reproduced from Reference [11] with permission from Elsevier.

**Figure 6 ijms-23-02195-f006:**
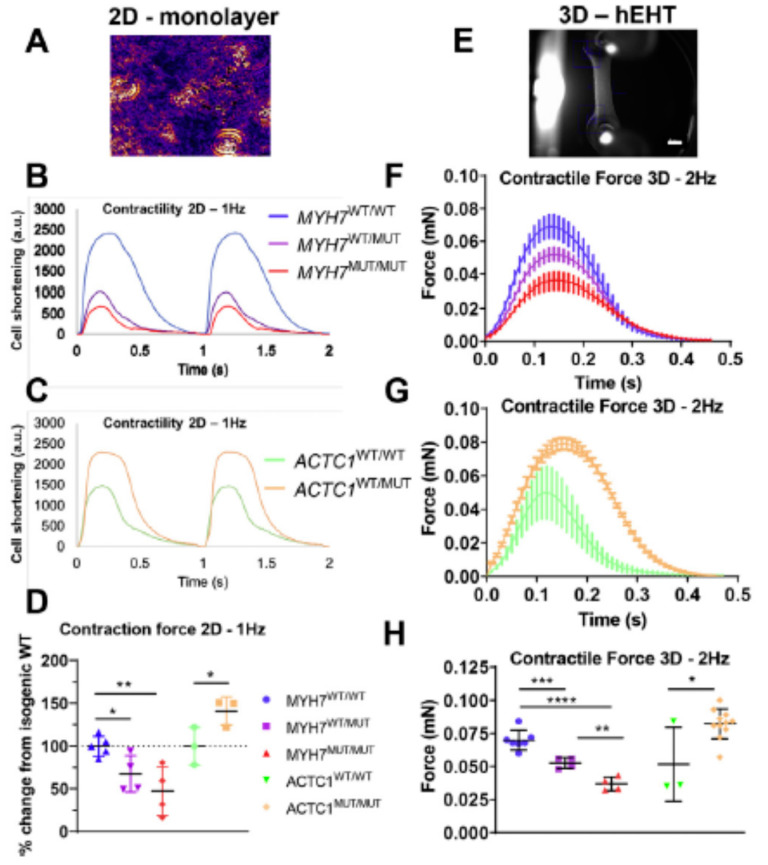
Results from experiments using hiPSC based preparations, illustrating both hypo- and hypercontractility in response to HCM-causing mutations and greater effects with homozygous mutations. (**A**) Micrograph illustrating hiPSC-derived cardiomyocytes of the type used for analyses in panels (**B**,**C**). (**B**) Representative recordings of cardiomyocyte shortening showing a hypocontractile phenotype for the β-MHC R453C mutation (MUT) compared to isogenic wild-type (WT) controls. Note that the hypocontractility was more severe with mutation homozygosity (MUT/MUT). (**C**) Similar recordings as in (B), showing a hypercontractile phenotype for the actin E99K heterozygous mutant. (**D**) Quantitative data for maximum cell shortening, as exemplified by recordings in (**B**,**C**), from 3–5 independent biological replicates. (**E**) Engineered heart tissue (hEHT) attached to silicon posts (scale bar = 1 mm) used for results in (**F**–**H**). (**F**) Twitch time courses (mean ± standard deviation) recorded from hEHTs, using electrical stimulation at 2 Hz. The hEHT contained engineered cardiomyocytes with β-MHC R453C mutation (heterozygous = WT/MUT; homozygous = MUT/MUT) or isogenic wild-type control cardiomyocytes (WT/WT). Note the hypocontractile phenotype with mutation. (**G**) Twitch time courses as in (**F**) but for hEHT heterozygous for the actin E99K mutation or with isogenic wild-type cardiomyocytes showing a hypercontractile phenotype. (**H**) Quantitative data for maximum hEHT force development, as exemplified by recordings in (**F**,**G**), from 3 to 11 independent biological replicates. Asterisks in (**D**,**H**) refer to statistical significance levels: * *p* < 0.05, ** *p* < 0.01, *** *p* < 0.001, **** *p* < 0.0001. Reproduced from Reference [32], under a CC-BY license (http://creativecommons.org/licenses/BY/4.0/) accessed on 18 January 2022.

**Figure 7 ijms-23-02195-f007:**
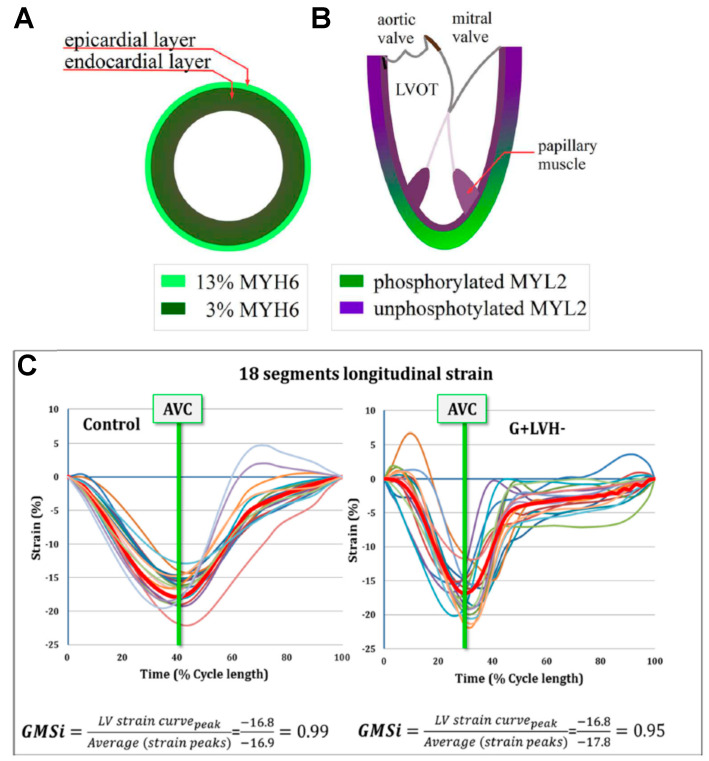
Macroscopic contractile non-uniformities between different parts of the left ventricle. (**A**) Percentage expression of the fast α-myosin (MYH6 gene) out of all myosin (α-myosin and slow β-myosin) varies between the subendocardial and subepicardial layers of the myocardial wall. (**B**) RLC (gene MYL2) phosphorylation level exhibits a basal–apical gradient leading to important effects on ventricular and valvular mechanics [35,204]. (**C**) Local longitudinal strain (differently colored traces; segment shortening, negative; lengthening, positive) in different parts of the ventricular wall during systole in genotype negative (control) individuals and in mutation carriers (G+, LVH−). Note the different behavior in mutation carriers compared to controls with greater non-uniformities during the early phases of systole, including noticeable elongation of some segments. AVC: aortic valve closure. GMSi: global mechanical synchrony index. GMSi < 1.0 suggests dyssynchrony. Panels (**A**,**B**) reproduced from Reference [35] under license CC-BY. Panels (**C**) reproduced from Reference [31], with permission from Elsevier.

**Figure 8 ijms-23-02195-f008:**
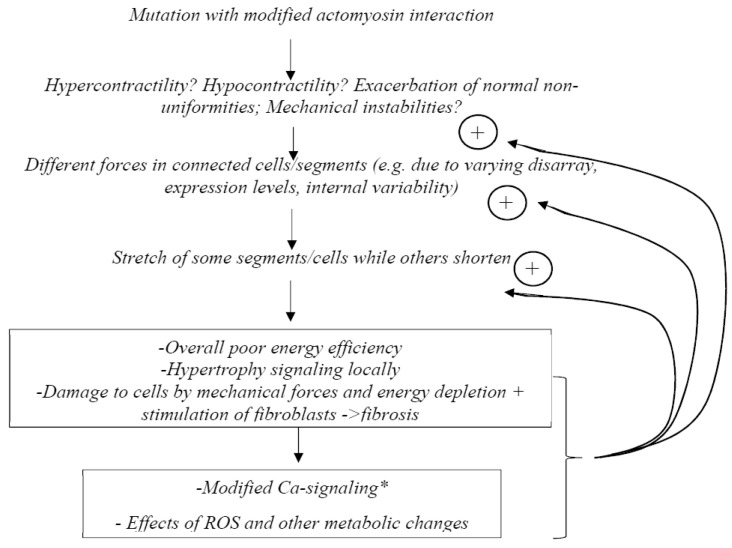
A possible unified model for pathogenesis in HCM. * Could also be a more upstream effect in the case of mutations that change Ca^2+^ regulation of the thin filament regulatory system.

**Table 1 ijms-23-02195-t001:** Three main hypotheses for primary effects of HCM mutations (before pathologic remodeling) in focus in the present paper.

Hypothesis	Main Characteristics	Key Papers
1. Hypercontractility	High power output, high force and velocity, high ATP turnover rate, high calcium sensitivity and diastolic dysfunction	[11,14,15,16,25,26,27]
2. Hypocontractility	Largely opposite to hypercontractility	[11,28,29,30,31,32]
3. Non-uniformity	Non-uniform contractile strength and/or instabilities along cardiac cells/fibers and between different parts of ventricular wall	[28,31,33,34,35,36]

**Table 2 ijms-23-02195-t002:** Hypercontractility vs. hypocontractility from experiments and analyses on different hierarchical levels (2011 and later) ^a^.

Expt. System	Isolated Proteins with One-Headed Myosin Fragments (S1)	Isolated Proteins Including Full Length Myosin or HMM-Like Constructs	Myofibrils (and to Limited Extent, Skinned Muscle Cells)		hiPSC	Whole Hearts in Patients Relying on Cardiac Imaging	
**Studied** **Parameter**	*Mutation carriers; patients; expressed proteins*	*Mutation carriers; patients, expressed proteins*	*Isolated from patients*	*Mutation carriers (reconstituted I-bands)*		*Patients*	*Carrier (no hypertrophy, no fibrosis)*
**Max force, power, velocity and ejection fraction (EF).**	Often **increased** but sometimes decreased (e.g., References [18,127]).	Generally **increased** but **not always** (e.g., see References [17,162]). Most HCM mutations (but not all) affect parts of β- MHC that take part in IHM.	Generally **decreased isometric force at saturating [Ca^2+^]** (reviewed in Reference [48]).	Purified or expressed thin filament proteins reconstituted, e.g., into bovine cardiac muscle give **variable effects on maximum force** [51,163,164,165].	Generally **increased**, but not always (e.g., Reference [32]).	Often **little/no effect** on EF and global systolic function, but other measures suggest **systolic dysfunction**.	Often **only minor effects** on EF and global systolic function, but evidence, e.g., from 2D strain imaging, for **subtle systolic dysfunction** [30,31,137,166].
**Energy depletion, inefficient energy usage**.	Often **unchanged or reduced ATP turnover rate with acto-subfragment 1 (e.g., References** [18,99,127,162]**),** but **increased ATP turnover in some early onset β-MHC mutations** [156].	Destabilized IHM and increased Ca-sensitivity suggest **increased ATP consumption**, not the least, under nearly relaxing conditions.	Evidence for **reduced efficiency (increased tension cost).**		Evidence for **reduced thermodynamic efficiency** and accumulation of metabolites and, in some cases (but not always), ROS ^b^.	Evidence for **energy depletion, accumulation of ROS and mitochondrial dysfunction** [154].	**Evidence for energy depletion (e.g., increased [PCr]/[ATP] ratio and increased oxygen consumption)** that occurs before evidence for diastolic dysfunction [80,167,168].
**Slowed relaxation, increased Ca-sensitivity (and related), diastolic dysfunction.**	**Increased Ca-sensitivity** [155,169] and **uncoupling of lusitropic effect of troponin I phosphorylation** [170] Data from in vitro motility assays and ATPase assays.		Generally **increased Ca-sensitivity** but **sometimes reduced** [28]**. Increase restored to normal by restored TnI phosphorylation** [48,171]**. Faster relaxation and faster cross-bridge detachment at full activation** [21,48].	Thin filament proteins reconstituted into cardiac muscle **generally give increased force at low [Ca]**, but in some cases reduced Ca sensitivity [164], as also seen with human RLC mutation in transgenic mice [172].	Generally **increased Ca-sensitivity and slowed relaxation.** In **addition, disturbed Ca-handling in some cases,** e.g., see Reference [32].	**Diastolic dysfunction**	Generally, **diastolic dysfunction** with **slowed relaxation** and compromised ventricular filling (e.g., reviewed in Reference [173]), **but not always** found [173].

^a^ Only a few references are given in the table (see text for more references), particularly references that contradict the dominant findings of hypercontractility in different forms. ^b^ ROS: Reactive oxygen species.

**Table 3 ijms-23-02195-t003:** Normal differences in force between cells being enhanced by 50% overall increase in force-development.

	Cell 1	Cell 2	Difference in Force
Normal (force)	1	1.2	0.2
Hypercontractility (50% increase in force per cell) in HCM but no other change	1.5	1.8	0.3

## Data Availability

Data is contained within the article.

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
