# Peer review of "Critical Evaluation of Current Hypotheses for the Pathogenesis of Hypertrophic Cardiomyopathy"

_ijms, 2022, doi:10.3390/ijms23042195_

Round 1
Reviewer 1 Report
The Authors presented the review paper "Critical evaluation of current hypotheses for the pathogenesis of hypertrophic cardiomyopathy ".
1) Some more 2-3 year references have to be presented in the introduction part to present the area perspectives.
2) It will be excellent to summarize the hypothesis in the introduction part in one Table with references and some argumentation.
3) Biochemistry scheme for calcium metabolism with dysfunction mechanism is required for section 2.4.
4) Section 2.5 is limited. It is very difficult to understand its necessity. I recommend enlarging it.
Reviewer 2 Report
The authors provided a huge manuscript on hypertrophic Cardiomyopathy. There are no real flaws that need to be addressed but the manuscript is very long, not easy to read because there are too many information. It is more likely to be considered as a book chapter than a review article. I suggest to be more concise and straight to the point.
Round 2
Reviewer 2 Report
The manuscript is upgraded in this version despite it remains very long.